# COMPOSE AND CONQUER: DIFFUSION-BASED 3D DEPTH AWARE COMPOSABLE IMAGE SYNTHESIS

**Jonghyun Lee**[1,2*]**, Hansam Cho**[1,2]**, Youngjoon Yoo**[2]**, Seoung Bum Kim**[1†]**, Yonghyun Jeong**[2†]
[1]Korea University, [2]NAVER Cloud
`{tomtom1103, chosam95, sbkim1}@korea.ac.kr`
`{youngjoon.yoo,yonghyun.jeong}@navercorp.com`

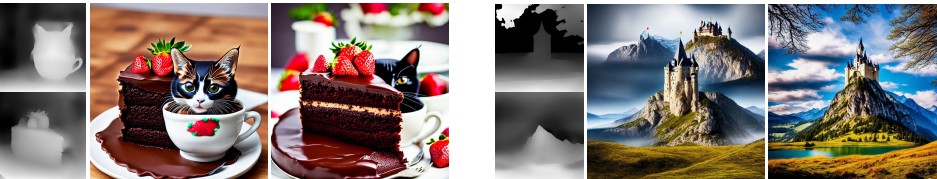

**(a) Foreground/Background object ordering.** Samples are generated via two depth maps (top left, bottom left) of foreground and background objects, with each object faithfully occluding each other.

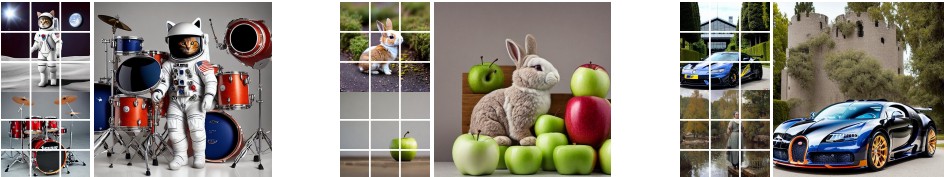

**(b) Localized global semantic injection.** Given two exemplar images (top left, bottom left), semantics of each image are injected onto a localized area in a disentangled manner.

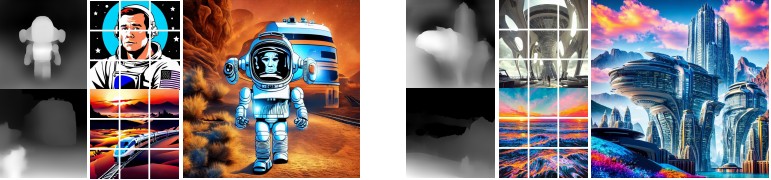

**(c) Composition with Foreground/Background depth maps and global semantics.** By mix & matching depth maps and exemplar images, CnC generates samples with localized local & global conditions.

Figure 1: Compose and Conquer is able to localize both local and global conditions in a 3D depth aware manner. For details on the figure, see Section 1.

## ABSTRACT

Addressing the limitations of text as a source of accurate layout representation in text-conditional diffusion models, many works incorporate additional signals to condition certain attributes within a generated image. Although successful, previous works do not account for the specific localization of said attributes extended into the three dimensional plane. In this context, we present a conditional diffusion model that integrates control over three-dimensional object placement with disentangled representations of global stylistic semantics from multiple exemplar images. Specifically, we first introduce *depth disentanglement training* to leverage the relative depth of objects as an estimator, allowing the model to identify the absolute positions of unseen objects through the use of synthetic image triplets. We also introduce *soft guidance*, a method for imposing global semantics onto targeted regions without the use of any additional localization cues. Our integrated framework, COMPOSE AND CONQUER (CNC), unifies these techniques to localize multiple conditions in a disentangled manner. We demonstrate that our approach allows perception of objects at varying depths while offering a versatile framework for composing localized objects with different global semantics.

---

[*]First Author. Work done during an internship at NAVER Cloud.
[†]Corresponding Co-Authors.

# 1 INTRODUCTION

Following the recent progress in text-conditional diffusion models (Rombach et al., 2022; Ramesh et al., 2022; Saharia et al., 2022; Nichol et al., 2021), many subsequent studies have emerged to address their inherent limitation in accurately representing the global layout of generated images. These follow-up works enrich the text-based conditioning capabilities of diffusion models by incorporating additional conditions such as segmentation maps (Zeng et al., 2023; Goel et al., 2023), depth maps (Zhang & Agrawala, 2023; Mou et al., 2023), bounding boxes (Li et al., 2023), and inpainting masks (Yang et al., 2023). These modifications effectively retain the extensive knowledge encapsulated in the pretrained priors.

Despite these advancements, two primary challenges persist in the current literature. Firstly, while existing models are efficient in generating an object under locally constrained conditions like depth maps and bounding boxes, which inherently capture structural attributes, they confine the generative space to a two-dimensional plane. This limitation makes them less adept at handling object placement within a three-dimensional (3D) or z-axis (depth) perspective, and hence vulnerable to generating images without properly reflecting the depth-aware placement of multiple objects. Secondly, the issue of applying global conditions, such as style and semantics, from multiple image sources to specific regions of the target image in a controlled manner has yet to be resolved.

To address the existing limitations on local and global conditions and also enhance the capabilities of image generation models, we introduce COMPOSE AND CONQUER (CnC). Our proposed CnC consists of two building blocks: a local fuser and global fuser designed to tackle each problem. First, the local fuser operates with a new training paradigm called *depth disentanglement training* (DDT) to let our model understand how multiple objects should be placed in relation to each other in a 3D space. DDT distills information about the relative placement of salient objects to the local fuser by extracting depth maps from synthetic image triplets, originally introduced in the field of image composition. Second, the global fuser employs a method termed *soft guidance*, which aids our model in localizing global conditions without any explicit structural signals. Soft guidance selectively masks out regions of the similarity matrix of cross-attention layers that attend to the specific regions of each salient object.

Figure 1 demonstrates the main capabilities of our model trained on DDT and soft guidance. In Figure 1(a), DDT lets our model infer relative depth associations of multiple objects within one image, and generates objects that are placed in different depths of the z-axis with foreground objects effectively occluding other objects. In Figure 1(b), we show that by applying soft guidance, our model can localize global semantics in a disentangled manner. By utilizing the local and global fuser simultaneously as demonstrated in Figure 1(c), our model gives users the ability to compose multiple localized objects with different global semantics injected into each localized area, providing a vast degree of creative freedom.

We quantitatively evaluate our model against other baseline models and gauge the fidelity of samples and robustness to multiple input conditions, and demonstrate that our model substantially outperforms other models on various metrics. We also evaluate our model in terms of reconstruction ability, and the ability to *order* objects into different relative depths. We shed light onto the use of DDT, where we demonstrate that DDT dissipates the need to provide additional viewpoints of a scene to infer the relative depth placement of objects. Furthermore, we show that soft guidance not only enables our model to inject global semantics onto localized areas, but also prevents different semantics from bleeding into other regions.

Our contributions are summarized as follows:

- We propose *depth disentanglement training* (DDT), a new training paradigm that facilitates a model's understanding of the 3D relative positioning of multiple objects.
- We introduce *soft guidance*, a technique that allows for the localization of global conditions without requiring explicit structural cues, thereby providing a unique mechanism for imposing global semantics onto specific image regions.
- By combining these two propositions, we present COMPOSE AND CONQUER (CnC), a framework that augments text-conditional diffusion models with enhanced control over three-dimensional object placement and injection of global semantics onto localized regions.

## 2 RELATED WORK

**Conditional Diffusion Models.** Diffusion models (DMs) (Sohl-Dickstein et al., 2015; Ho et al., 2020) are generative latent variable models that are trained to reverse a forward process that gradually transforms a target data distribution into a known prior. Proving highly effective in its ability to generate samples in an unconditional manner, many following works (Dhariwal & Nichol, 2021; Ho et al., 2022; Nichol et al., 2021; Rombach et al., 2022; Ramesh et al., 2022) formulate the diffusion process to take in a specific condition to generate corresponding images. Out of said models, Rombach et al. (2022) proposes LDM, a latent text-conditional DM that utilizes an autoencoder, effectively reducing the computational complexity of generation while achieving high-fidelity results. LDMs, more commonly known as Stable Diffusion, is one of the most potent diffusion models open to the research community. LDMs utilize a twofold approach, where an encoder maps $\mathbf{x}$ to its latent representation $\mathbf{z}$, and proceeds denoising $\mathbf{z}$ in a much lower, memory-friendly dimension. Once fully denoised, a decoder maps $\mathbf{z}$ to the original image dimension, effectively generating a sample.

**Beyond Text Conditions.** While text-conditional DMs enable creatives to use free-form prompts, text as the sole condition has limitations. Namely, text-conditional DMs struggles with localizing objects and certain semantic concepts with text alone, because text prompts of large web-scale datasets used to train said models (Schuhmann et al., 2021) do not provide explicit localized descriptions and/or semantic information. Addressing this limitation, many works have introduced methods to incorporate additional conditional signals to the models while preserving its powerful prior, *e.g.* freezing the model while training an additional module. Among these models, ControlNet (Zhang & Agrawala, 2023) and T2I-Adapter (Mou et al., 2023) train additional modules that incorporate modalities such as depth maps and canny edge images to aid generation of localized objects. However, these models only support a single condition, lacking the ability to condition multiple signals or objects. Taking inspiration from ControlNet, Uni-ControlNet (Zhao et al., 2023) extends its framework to accept multiple local conditions and a single global condition at once. Whereas the works detailed above all leverage Stable Diffusion as a source of their priors, Composer (Huang et al., 2023) operates in the pixel-space. Although being able to process multiple conditions at once, both Composer and Uni-ControlNet struggle in processing incompatible conditions, or conditions that overlap with each other. They also do not provide methods to localize global semantics onto a localized region. In contrast, our approach directly addresses these challenges by proposing two novel methods, depth disentanglement training and soft guidance, which enables the composition of multiple local/global conditions onto localized regions.

## 3 METHODOLOGY: COMPOSE AND CONQUER

The architecture illustrated in Figure 2 shows the overall framework of our proposed method. CnC consists of a local fuser, a global fuser, and components of a pretrained text-conditional DM. Our local fuser captures the relative z-axis placements of images through depth maps, and our global fuser imposes global semantics from CLIP image embeddings (Radford et al., 2021) on specified regions. Explanations of the local and global fuser are detailed below.

### 3.1 GENERATIVE PRIOR UTILIZATION

In line with earlier studies that incorporate additional condition signals (Li et al., 2023; Mou et al., 2023; Zhang & Agrawala, 2023; Zhao et al., 2023), we use the LDM variant known as Stable Diffusion (SD) as our source of prior. Specifically, SD utilizes a UNet (Ronneberger et al., 2015) like structure, where noisy latent features consecutively pass through 12 spatial downsampling blocks, one center block $C$, and 12 spatial upsampling blocks. Each block consists of either a ResNet (He et al., 2016) block or Transformer (Vaswani et al., 2017) block, and for brevity, we respectively refer to each group of the 12 blocks as the encoder $E$ and decoder $D$. Inspired by ControlNet Zhang & Agrawala (2023) and Uni-ControlNet Zhao et al. (2023), the Stable Diffusion architecture is utilized twofold in our model, where we first freeze the full model and clone a trainable copy of the encoder and center block, denoted $E'$ and $C'$. We initialize the weights of the full model from SD, and the weights of $E'$ and $C'$ from Uni-ControlNet. The cloned encoder $E'$ and center block $C'$ receives localized signals from our local fuser, which acts as the starting point of our model. We detail our model architecture, methodologies of the two building blocks and corresponding training paradigms in the section below.

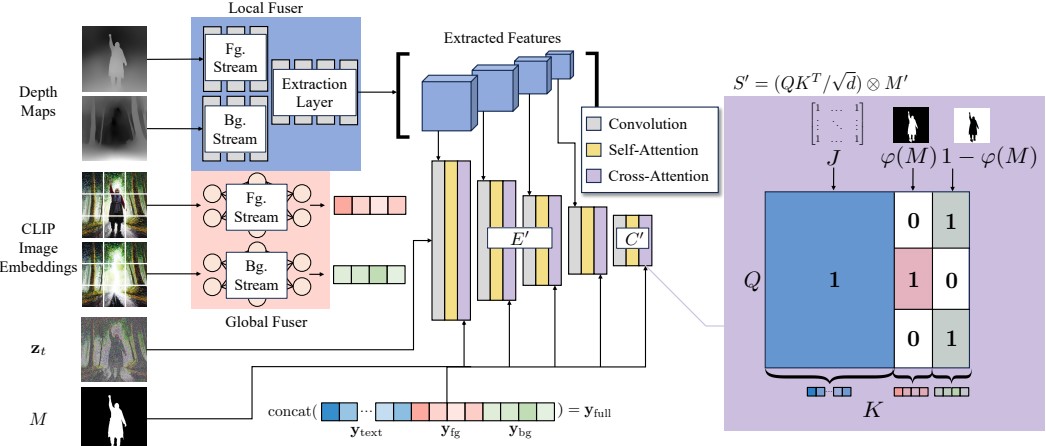

Figure 2: **Model Architecture**. Our model consists of a local fuser, a global fuser, and the cloned encoder/center block $\{E', C'\}$. The input depth maps are fed into the local fuser, producing four latent representations of different spatial resolutions, which are incorporated into $E'$. The CLIP image embeddings are fed into the global fuser, producing 2 extra tokens to be concatenated with the text token embeddings. Masks $M$ are flattened and repeated to produce $M' = \text{concat}(J, \varphi(M), 1 - \varphi(M))$, which serves as a source of soft guidance of the cross-attention layers.

## 3.2 LOCAL FUSER

We first provide details on our local fuser, which incorporates depth maps extracted form a pretrained monocular depth estimation network (Ranftl et al., 2020) as our local condition. Specifically, our local fuser serves as a source of localized signals that are incorporated into the frozen SD blocks. We also provide details on *depth disentanglement training*, and how synthetic image triplets are leveraged as sources of relative depth placements of objects.

**Synthetic Image Triplets.** For a model to be able to represent overlapping objects with varied scopes of depth during inference, the model needs to learn to recognize different elements obscured by objects during training. Although straightforward in a 3D world, informing a model about objects that are occluded by another in a 2D image is non-trivial, due to the fact that once the image is captured, any spatial information about objects behind it is forever lost. To overcome this limitation, we first adopt a process utilized in image composition (Fang et al., 2019) to generate synthetic image triplets, which serves as training samples for our depth disentanglement training (DDT), detailed in the next section. The synthetic image triplets $\{I_f, I_b, M\}$ are derived from a single source image $I_s \in \mathbb{R}^{H \times W \times 3}$, and is composed of the foreground image $I_f \in \mathbb{R}^{H \times W \times 3}$, the background image $I_b \in \mathbb{R}^{H \times W \times 3}$, and a binary foreground object mask $M \in \{0,1\}^{H \times W}$. The foreground image $I_f$ is derived using the Hadamard product of $I_f = I_s \otimes M$, leaving just the salient object of $I_s$. To generate $I_b$, we utilize Stable Diffusion's inpainting module (Rombach et al., 2022). This is achieved by inpainting the result of $I_s \otimes (1 - \tilde{M})$, where $\tilde{M}$ is the binary dilated $M$. Conceptually, this process can be thought of as inpainting the depiction of $I_s$ without its salient object, effectively letting our model *see* behind it. For details on this choice, see appendix A.2.

**Depth Disentanglement Training.** Once the synthetic image triplets $\{I_f, I_b, M\}$ are prepared, we proceed to extract the depth maps of $I_f$ and $I_b$ to train our local fuser, which we refer to as depth disentanglement training (DDT). Our local fuser incorporates these two depth maps, which passes through its own individual stream consisting of ResNet blocks, and are concatenated along its channel dimension. Apart from previous works that directly fuse different local conditions before entering a network, DDT first process each depth map of $I_f$ and $I_b$ in their own independent layers. Reminiscent of early and late fusion methodologies of salient object detection (Zhou et al., 2021), we consider DDT a variant of late fusion, where the network first distinguishes each representation in a disentangled manner. Once concatenated, features containing spatial information about objects

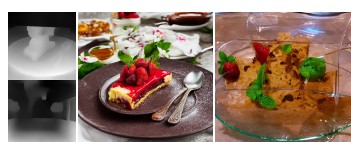 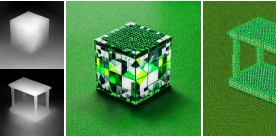 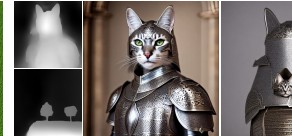

(a) A beautiful dessert waiting to be shared by two people   (b) Diamond Minecraft block facing frontwards on a green background   (c) Cat knight, portrait, finely detailed armor, intricate design, silver, silk, cinematic lighting

Figure 3: **Depth disentanglement training.** Our model trained on DDT **(Left)** successfully recognizes that objects portrayed by the foreground depth map **(Top left)** should be placed closer that the background depth map **(Bottom left)**, and fully occludes objects that are larger. On the other hand, when trained on just the depth maps of $I_s$ **(Right)**, our model struggles to disentangle the depth maps, resulting in either objects becoming fused (samples (a), (c)) or completely ignoring the foreground object (sample (b)).

in varying depths are extracted along different resolutions by an extraction layer. These features are then incorporated into the cloned and frozen SD blocks, which we provide additional details in our appendix A.2. We formulate DDT to train our model because to represent overlapping objects with varied depths during inference, the model needs to recognize the elements obscured by the salient object. By providing our model with an explicit depth representation of what lies behind the salient object—albeit synthetic—, our model is able to effectively distinguish the relative depths of multiple objects. Figure 3 demonstrates the effect of training our model with DDT compared to training our model on depth maps of $I_s$, as done in previous works. Even though that our local fuser was trained on depth maps that convey only the relative depth associations between the foreground salient object and background depth maps, we find that our model extends to localizing different salient objects.

## 3.3 GLOBAL FUSER

While our local fuser incorporates depth maps as a source of relative object placements, our global fuser leverages *soft guidance* to localize global semantics onto specific regions. We use image embeddings derived from the CLIP image encoder (Radford et al., 2021) as our global semantic condition. This choice is informed by the training methodology of SD, which is designed to incorporate text embeddings from its contrastively trained counterpart, the CLIP text encoder. The text embeddings $\mathbf{y}_{\text{text}}$ are integrated into the intermediate SD blocks through a cross-attention mechanism. In this setup, the text embeddings serve as the context for keys and values, while the intermediate noised latents act as the queries. Although prior works (Nichol et al., 2021; Ramesh et al., 2022; Huang et al., 2023) have merged CLIP image embeddings with text embeddings within the cross-attention layers of the DM, this approach lacks the provision of spatial grounding information. As a result, the global semantics are conditioned on the entire generated image, lacking precise localization capabilities. To overcome this limitation, our method leverages the binary foreground mask $M$ used to extract $I_f$ in our synthetic image triplet.

**Soft Guidance.** In detail, we first project the image embeddings of $I_s$ and $I_b$ using our global fuser, which consists of stacked feedforward layers. The global fuser consists of separate foreground/background streams, where each image embedding is projected and reshaped to $N$ global tokens each, resulting in $\mathbf{y}_{\text{fg}}$ and $\mathbf{y}_{\text{bg}}$. Unlike our local fuser module, our global fuser doesn't fuse each stream within the feedforward layers. We instead choose to concatenate $\mathbf{y}_{\text{fg}}$ and $\mathbf{y}_{\text{bg}}$ directly to $\mathbf{y}_{\text{text}}$, where the extended context $\mathbf{y}_{\text{full}} = \text{concat}(\mathbf{y}_{\text{text}}, \lambda_{\text{fg}}\mathbf{y}_{\text{fg}}, \lambda_{\text{bg}}\mathbf{y}_{\text{bg}})$ is utilized in the cross-attention layers of our cloned and frozen modules. $\lambda_{\text{fg}}$ and $\lambda_{\text{bg}}$ denotes scalar hyperparameters that control the weight of each token, which are set to 1 during training. In the cross-attention layers, the similarity matrix $S$ is given by $S = (QK^T/\sqrt{d})$ with $Q = W_Q \cdot \mathbf{z}_t$ and $K = W_K \cdot \mathbf{y}_{\text{full}}$, where $\mathbf{z}_t$ is a noised variant of $\mathbf{z}$ with the diffusion forward process applied $t$ steps.

Once $S$ is calculated, we apply soft guidance by first creating a Boolean matrix $M'$, which has the same dimensionality of $S$. Given that $S \in \mathbb{R}^{i \times j}$, $M'$ is defined as $M' = \text{concat}(J, \varphi(M), 1 - \varphi(M))$, where $J \in \mathbf{1}^{i \times j - 2N}$ denotes an all ones matrix, $\varphi(M) \in \mathbb{B}^{i \times N}$ denotes the reshaped, flattened and repeated boolean mask $M$, and $1 - \varphi(M)$ denotes the complement of $\varphi(M)$ of the same shape. By overriding $S$ with the Hadamard product $S' = S \otimes M'$, the attention operation $\text{softmax}(S') \cdot V$ is completed. Intuitively, soft guidance can be thought of as masking out parts of

$S$ where $\mathbf{z}_t$ should not be attending to, *e.g.* forcing the cross-attention computations of $\mathbf{y}_{\text{fg}}$ and $\mathbf{y}_{\text{bg}}$ to be performed only on its corresponding flattened values of $\mathbf{z}_t$. We find that by letting the tokens of $\mathbf{y}_{\text{text}}$ attend to the whole latent and restricting the extra tokens, generated samples are able to stay true to their text conditions while also reflecting the conditioned global semantics in their localized areas, even though that spatial information of $M$ is forfeited.

## 3.4 Training

LDMs are optimized in a noise-prediction manner, utilizing a variation of the reweighted variational lower bound on $q(\mathbf{x}_0)$, first proposed by Ho et al. (2020). We extend this formulation and optimize our model with Eq. 1 to learn the conditional distribution of $p(\mathbf{z}|y)$, where $y$ denotes the set of our conditions of depthmaps and image embeddings accompanied with CLIP text embeddings. As mentioned above, we freeze the initial SD model while jointly optimizing the weights of the cloned encoder $E'$, center block $C'$, and the local/global fuser modules, denoted as $\theta'$.

$$\min_{\boldsymbol{\theta'}} \mathcal{L} = \mathbb{E}_{\mathbf{z}, \boldsymbol{\epsilon} \sim \mathcal{N}(\mathbf{0}, \mathbf{I}), t} \left[ \left\| \boldsymbol{\epsilon} - \boldsymbol{\epsilon}_{\{\boldsymbol{\theta}, \boldsymbol{\theta'}\}} (\mathbf{z}_t, t, y) \right\|_2^2 \right] \tag{1}$$

## 4 Experiments

### 4.1 Experimental Setup

**Datasets.** Our synthetic image triplets $I_f, I_b, M$ are generated from two distinct datasets: COCO-Stuff (Caesar et al., 2018) and Pick-a-Pic (Kirstain et al., 2023). We refer the readers to Section 4.3 for our reasoning behind this choice. The COCO-Stuff dataset, with 164K images, has pixel-wise annotations classifying objects into "things" (well-defined shapes) and "stuff" (background regions). We leverage the fact that objects in the things category can be seen as the salient object, and create $M$ by setting each pixel of the indexed mask to 1 if it belongs to either one of the 80 things classes, and 0 otherwise. Text prompts are randomly chosen from five available for each image. The Pick-a-Pic dataset contains 584K synthetic image-text pairs generated by SD and its variants, and was collected as an effort to train a preference scoring model. Each text prompt is paired with two generated images, and holds a label denoting the preferred image in terms of fidelity and semantic alignment to the given prompt. By only keep the preferred image and filtering out inappropriate content, we end up with 138K image-text pairs. Because Pick-a-Pic doesn't hold ground truth labels for the salient object unlike COCO-Stuff, we utilize a salient object detection module (Qin et al., 2020) to generate $M$. Combining these two datasets, we generate 302K synthetic image triplets $\{I_f, I_b, M\}$ through the process detailed in Section 3.2.

**Implementation Details.** For depth map representations, we utilize a monocular depth estimation network (Ranftl et al., 2020), and the CLIP image encoder (Radford et al., 2021) for global semantic conditions. Although formulated as a single model, we empirically find that training the local and global fuser independently, and finetuning the combined weights lead to faster convergence. During training, images are resized and center cropped to a resolution of $512 \times 512$. We train our local fuser with the cloned $E'$ and $C'$ for 28 epochs, our global fuser for 24 epochs, and finetune the full model for 9 epochs, all with a batch size of 32 across 8 NVIDIA V100s. During training, we set an independent dropout probability for each condition to ensure that our model learns to generalize various combinations. For our evaluation, we employ DDIM (Song et al., 2020) sampling with 50 steps, and a CFG (Ho & Salimans, 2021) scale of 7 to generate images of $768 \times 768$.

### 4.2 Evaluation

**Qualitative Evaluation.** We demonstrate the results of our method compared to other baseline models that incorporate either depth maps as a local condition, CLIP image embeddings as a global condition, or both. The baseline models are listed in Table 1. GLIGEN (Li et al., 2023) and ControlNet (Zhang & Agrawala, 2023) are trained to accept a single depth map as a local condition. Uni-ControlNet (Zhao et al., 2023) and T2I-Adapter (Mou et al., 2023) are trained to accept a depth map and an exemplar image as a source of CLIP image embeddings. In Figure 4, we show qualitative results of our model compared to models that accept both depth maps and exemplar images as a source of global semantics. Since our model accepts two depth maps and two exemplar images, note that we condition the same images for each modality in Figure 4. While other models

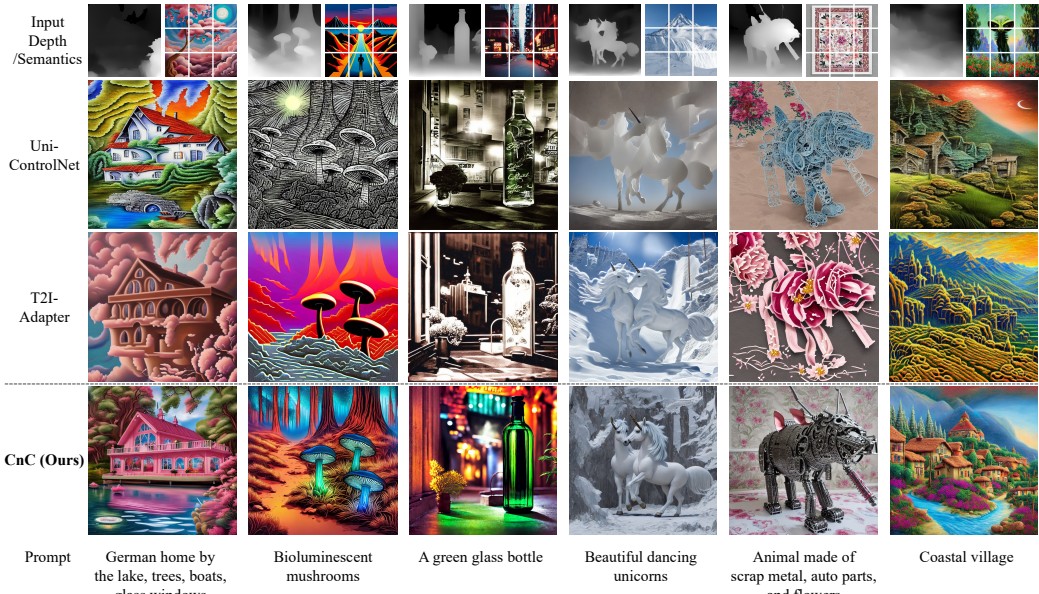

Figure 4: **Samples compared to other baseline models.** Compared to others, CnC strikes a balance between the given depth maps, exemplar images, and text prompts.

| Model/Condition | Depth | | | Semantics | | | Depth+Semantics | | |
|---|---|---|---|---|---|---|---|---|---|
| | FID (↓) | IS (↑) | CLIP Score (↑) | FID (↓) | IS (↑) | CLIP Score (↑) | FID (↓) | IS(↑) | CLIP Score (↑) |
| GLIGEN | 18.887 | 29.602 | 25.815 | - | - | - | - | - | - |
| ControlNet | **17.303** | **31.652** | 25.741 | - | - | - | - | - | - |
| Uni-ControlNet | 19.277 | 31.287 | 25.620 | 23.632 | 28.364 | 24.096 | 18.945 | 28.218 | 24.839 |
| T2I-Adapter | 20.949 | 31.485 | 26.736 | 35.812 | 23.254 | 23.666 | 30.611 | 23.938 | 24.579 |
| **CnC** | 19.804 | 27.555 | 25.211 | 35.178 | 21.932 | 22.161 | 21.318 | 25.421 | 24.659 |
| **CnC Finetuned** | 22.257 | 27.981 | **26.870** | **17.254** | **32.131** | **25.940** | **18.191** | **29.304** | **25.880** |

Table 1: Evaluation metrics on the COCO-Stuff val-set. We omit the results of semantics and depth+semantics on GLIGEN and ControlNet due to the models not supporting these conditions. Best results are in **bold**.

do grasp the ideas of each condition, it can be seen that our model exceeds in finding a balance between the structural information provided by the depth maps and semantic information provided by exemplar images and text prompts. Taking the first column as an example, Uni-ControlNet often fails to incorporate the global semantics, while T2I-Adapter often overconditions the semantics from the exemplar image, ignoring textual cues such as "lake" or "boats". Our approach adeptly interweaves these aspects, accurately reflecting global semantics while also emphasizing text-driven details and structural information provided by depth maps. For additional qualitative results, we refer the readers to Figure 5.

**Quantitative Evaluation.** As our baseline metrics, we utilize FID (Heusel et al., 2017) and Inception Score (Salimans et al., 2016) to evaluate the quality of generated images, and CLIPScore (Hessel et al., 2021) to evaluate the semantic alignment of the generated images to its text prompts. Table 1 reports the results evaluated on 5K images of the COCO-Stuff validation set. It can be seen that while our model with the local and global fuser trained independently and joined during inference (CnC, for brevity) shows adequate performance, our finetuned model excels in most metrics, except for FID and IS of our depth-only experiment. This may be attributed to the fact that while baseline models only take in a single depth map extracted from the validation image, our model takes in an additional depth map with inpainted regions that may not reflect the prior distribution of the dataset. The fact that our model has to process depth maps of conflicting localization information is another possible explanation. Though all models report similar CLIPScores, given their shared generative prior of SD, our finetuned model excels when generating from just an exemplar image, thanks to the integration of soft guidance. We conclude that our finetuned model achieves substantial performance

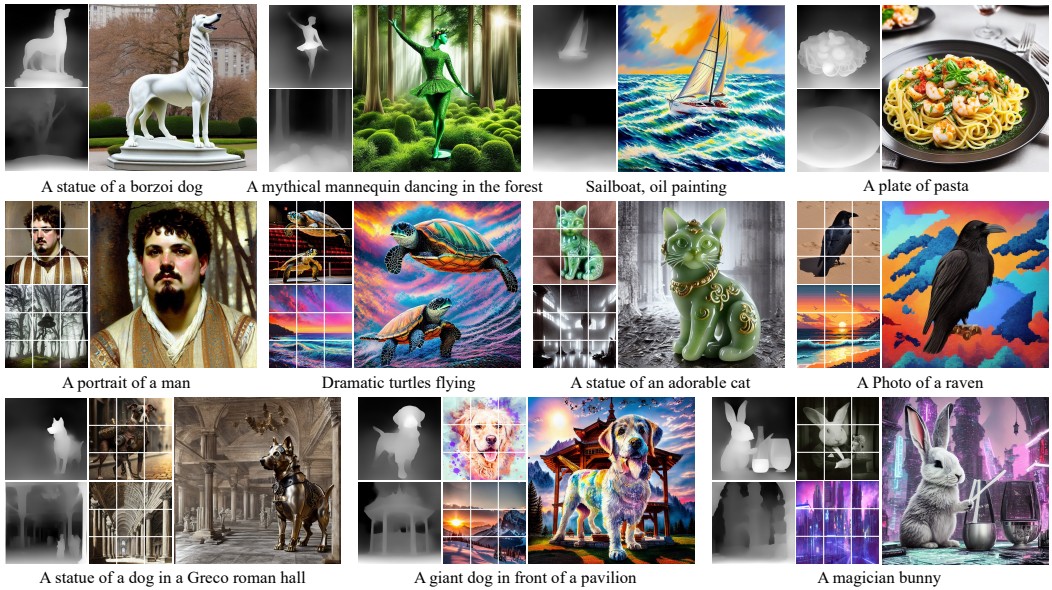

Figure 5: **Qualitative Results.** Foreground/background conditions are on the left of each sample.

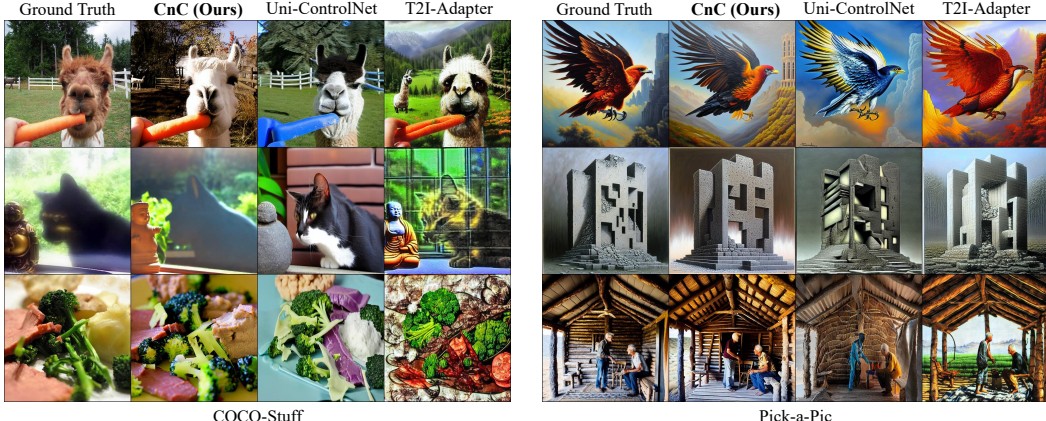

Figure 6: **Qualitative reconstruction comparison.** Samples are generated using conditions extracted from validation samples of COCO-Stuff (Left) and Pick-a-Pic (Right).

due to its additional finetuning phase, which enables the fusers to better adapt and understand each other's conditioning processes. See Section A.3 for results on the Pick-a-Pic validation set.

**Reconstruction.** We additionally report quantitative reconstruction metrics evaluated on COCO-Stuff and Pick-a-Pic validation sets, listed in Table 2. For reconstruction, our model utilizes the depth maps and CLIP image embeddings extracted from the image triplets of ground truth validation images, and baseline models utilize the depth maps and CLIP image embeddings extracted from the ground truth images, since they do not support more than one condition per modality. We adopt LPIPS (Zhang et al., 2018) as our metric of perceptual similarity and SSIM (Wang et al., 2004) as our metric of structural similarity. We also report the MAE of ground truth depth maps and the depth maps extracted from its generated counterpart as an additional measure of structural similarity, extended to the z-axis. Apart from the SSIM value of COCO-Stuff, we find that our model outperforms other models by a large margin. As seen in Figure 6, we find that our model is able to faithfully recreate objects in different localized areas while preserving its depth of field. While other baseline models succeed in localizing objects, they struggle in synthesizing the depth perspective, resulting in images looking relatively flat.

**Ablation Study.** The phenomenon known as concept bleeding (Podell et al., 2023) leads to different semantics to overlap with each other, resulting in unintentional results. Soft guidance allows

| Model/Condition | COCO-Stuff | | | Pick-a-Pic | | |
|---|---|---|---|---|---|---|
| | SSIM(↑) | LPIPS(↓) | MAE(↓) | SSIM(↑) | LPIPS(↓) | MAE(↓) |
| Uni-ControlNet | **0.2362** | 0.6539 | 0.1061 | 0.2506 | 0.6504 | 0.1111 |
| T2I-Adapter | 0.1907 | 0.6806 | 0.1201 | 0.2238 | 0.6724 | 0.1270 |
| **CnC** | 0.2345 | 0.6621 | 0.1061 | 0.2436 | 0.6431 | 0.1080 |
| **CnC Finetuned** | 0.2248 | **0.6509** | **0.0990** | **0.2690** | **0.6216** | **0.1027** |

Table 2: Quantitative reconstruction metrics evaluated on COCO-Stuff and Pick-a-Pic val-sets.

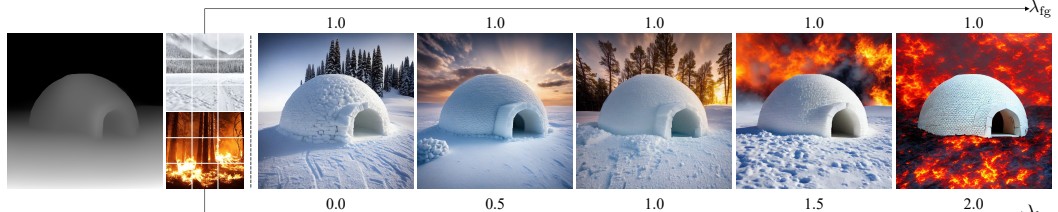

Figure 7: **The effect of soft guidance with conflicting semantics.** We condition the same depth map for each stream in the local fuser, and generate each sample with the prompt "An Igloo". By fixing $\lambda_{fg}$ and increasing $\lambda_{bg}$, the effect of the background global semantics substantially increases. Soft guidance prevents the two global semantics from bleeding into each other, *e.g.* concept bleeding, effectively maintaining the semantics of the igloo.

global semantics to be conditioned onto localized regions while preventing this undesirable effect. Figure 7 demonstrates this capability, where two contradicting semantics are localized. By fixing $\lambda_{fg}$ to 1 and steadily increasing $\lambda_{bg}$, the effects of the background semantics are amplified. However, due to soft guidance, even as the background semantics intensify, the contradicting semantic of the foreground object stays intact. Even though spatial information of $M$ is lost during soft guidance, we find that it faithfully creates a barrier for any semantics from bleeding in. For additional ablations, see Section A.4.

### 4.3 DISCUSSIONS

**Dataset Choices.** As mentioned in Section 4.1, the two datasets we employ to train our model are very different from another. Namely, images in COCO-Stuff include everyday scenes while images in Pick-a-Pic are fundamentally synthetic, being generated by variants of SD from prompts that transcend any description of real life scenarios. This design choice is intentional: we first point to the fact that most of our baseline models are trained on variants of MS-COCO (Lin et al., 2014). These models show that training only on real images as a method to introduce new conditions are adequate, but Kirstain et al. (2023) and Podell et al. (2023) report that COCO zero-shot FID is *negatively correlated* with human preferences and visual aesthetics of generated images. Although images from COCO and its variants do serve its purpose on introducing new conditions, we argue that leveraging another dataset that aligns with the learnt prior of pretrained DMs provides a safety net from prior drifting. By harnessing the detailed ground truth pixel-wise annotations of COCO-Stuff and letting our model learn additional representations from its original prior provided by Pick-a-Pic, we take advantage of the best of both worlds; providing a robust posterior of conditions while staying true to the user-preferred prior of DMs.

## 5 CONCLUSION & LIMITATIONS

We presented Compose and Conquer (CnC), a novel text-conditional diffusion model addressing two main challenges in the field: three-dimensional placement of multiple objects and region-specific localization of global semantics from multiple sources. CnC employs two main components: the local and global fuser, which respectively leverages the new Depth Disentanglement Training (DDT) and soft guidance techniques. We show that DDT infers the absolute depth placement of objects, and soft guidance is able to incorporate semantics on to localized regions. Evaluations on the COCO-stuff and Pick-a-Pic datasets illustrates CnC's proficiency in addressing these challenges, as demonstrated through extensive experimental results. Since the current framework limits the number of available conditions and the disentangled spatial grounds to the foreground and background, we leave the further decomposition of images into depth portraying primitives and the middle ground to leverage for future work.

## ACKNOWLEDGEMENTS

We thank the ImageVision team of NAVER Cloud for their thoughtful advice and discussions. Training and experiments were done on the Naver Smart Machine Learning (NSML) platform (Kim et al., 2018). This study was supported by BK21 FOUR.

## ETHICS STATEMENT

Diffusion models, as a type of generative model, have the potential to generate synthetic content that could be used in both beneficial and potentially harmful ways. While our work aims to advance the understanding and capabilities of these models, we acknowledge the importance of their responsible use. We encourage practitioners to consider the broader societal implications when deploying such models and to implement safeguards against malicious applications. Specifically, the diffusion model we utilize as a source of prior in our work is trained on the LAION (Schuhmann et al., 2021) dataset, a web-scraped collection. Despite the best intentions of the dataset's creators to filter out inappropriate data, LAION includes content that may be inappropriate for models to internalize, such as racial stereotypes, violence, and pornography. Recognizing these challenges, we underscore the necessity of rigorous scrutiny in using such models to prevent the perpetuation of harmful biases and misinformation.

## REPRODUCIBILITY STATEMENT

The source code and pretrained models can be found at https://github.com/tomtom1103/compose-and-conquer.

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

## A  APPENDIX

### A.1  EXTENDED RELATED WORK

**Image Composition.**  Image composition (Niu et al., 2021) involves the task of blending a given foreground with a background to produce a unified composite image. Traditional methods typically follow a sequential pipeline comprising of object placement, image blending/harmonization, and shadow generation. These steps aim to minimize the visual discrepancies between the two image components. With the recent advances in generative models, notably GANs Goodfellow et al. (2014) and DMs Ho et al. (2020), the image composition challenge has been reframed as a generative task. While GAN-based models have led in terms of the number of research contributions, diffusion-based models, as exemplified by works such as ObjectStitch Song et al. (2022) and Paint By Example Yang et al. (2023), showcase the potential of DMs as a one-shot solution for image composition, offering a departure from the multi-step traditional methods. However, it is essential to note that our approach diverges from typical image composition. Rather than aiming to preserve the distinct identity of the foreground and background, our model utilizes them as localized representations for text and global semantics to fill. Although our work aims to solve an inherently different task, we draw parallels to image compositioning in the way we leverage synthetic image triplets and handle the target image to be generated.

### A.2  DETAILS ON CNC

**Details on the Local Fuser.**  Depth disentanglement training (DDT) leverages synthetic image triplets $\{I_f, I_b, M\}$ in order to train the local fuser. DDT first incorporates the depth maps of $I_f$ and $I_b$ in their own foreground/background streams, as shown in Figure 2. The features from the streams are concatenated along their channel dimension, and features that incorporate both spatial features about $I_f$ and $I_b$ are extracted in four spatial resolutions. Each extracted feature subsequently passes through a zero convolution layer, and are finally incorporated into $E'$ through feature denormalization layers (Park et al., 2019), as done in Zhao et al. (2023). The frozen SD receives the localized signals from $E'$ and $C'$ at its decoder $D$, integrated by residual skip connections. Denoting outputs of the $i$-th blocks of $E$, $D$, and $C$ as $\mathbf{e}_i$, $\mathbf{d}_i$, and $\mathbf{c}$ respectively, and the corresponding outputs of $E'$ and $C'$ as $\mathbf{e}'_i$, and $\mathbf{c}'$, the integration is captured as:

$$\begin{cases} \text{concat}\left(\mathbf{c} + \mathbf{c}', \mathbf{e}_j + \mathbf{e}'_j\right) & \text{where } i = 1, \quad i + j = 13. \\ \text{concat}\left(\mathbf{d}_{-1}, \mathbf{e}_j + \mathbf{e}'_j\right) & \text{where } 2 \leq i \leq 12, \quad i + j = 13. \end{cases} \tag{2}$$

**Details on the Global Fuser.**  We provide an algorithm outlining the process of incorporating soft guidance to the cross-attention layers to train the global fuser below.

---

**Algorithm 1** Soft guidance for a single training timestep $t$

---

**Require:** $I_s$, $I_b$, $M$, $\mathbf{y}_{\text{text}}$, $\lambda_{\text{fg}} = 1$, $\lambda_{\text{bg}} = 1$, $\mathbf{z}_t$, CLIP IMAGE ENCODER, GLOBAL FUSER, $(E, C, D)$(Frozen layers of SD)

1: $(E_s, E_b) \leftarrow$ CLIP IMAGE ENCODER$(I_s, I_b)$
2: $(\mathbf{y}_{\text{fg}}, \mathbf{y}_{\text{bg}}) \leftarrow$ GLOBAL FUSER$(E_s, E_b)$
3: $\mathbf{y}_{\text{full}} \leftarrow \text{concat}(\mathbf{y}_{\text{text}}, \lambda_{\text{fg}}\mathbf{y}_{\text{fg}}, \lambda_{\text{bg}}\mathbf{y}_{\text{bg}})$
4: **for all** cross-attention layers in $E, C, D$ **do**
5: $\quad (Q, K, V) \leftarrow (W_Q \cdot \mathbf{z}_t, W_K \cdot \mathbf{y}_{\text{full}}, W_V \cdot \mathbf{y}_{\text{full}})$
6: $\quad S \leftarrow (QK^T/\sqrt{d})$ $\qquad\qquad\qquad\qquad\qquad\qquad\qquad\qquad\quad \triangleright S \in \mathbb{R}^{i \times j}$
7: $\quad J \leftarrow \mathbf{1}^{i \times (j-2N)}$ $\qquad\qquad\qquad\qquad\qquad\qquad \triangleright$ Initialize $J$ as an all ones matrix
8: $\quad \varphi(M) \leftarrow \text{Reshape}, \text{Flatten}, \text{Repeat}(M)$ $\qquad\qquad\qquad \triangleright \varphi(M) \in \mathbb{B}^{i \times N}$
9: $\quad M' \leftarrow \text{concat}(J, \varphi(M), 1 - \varphi(M))$
10: $\quad S' \leftarrow S \otimes M'$
11: $\quad \mathbf{z}_t \leftarrow \text{softmax}(S') \cdot V$
12: **end for**

---

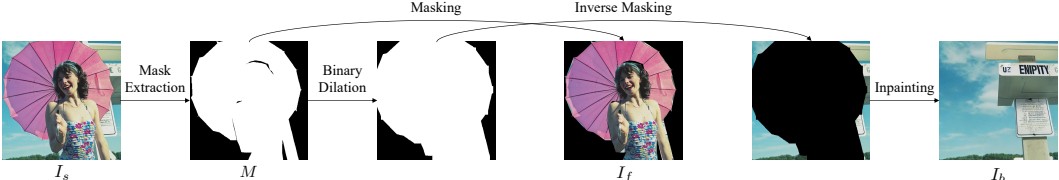

Figure 8: The process of generating our synthetic image triplets.

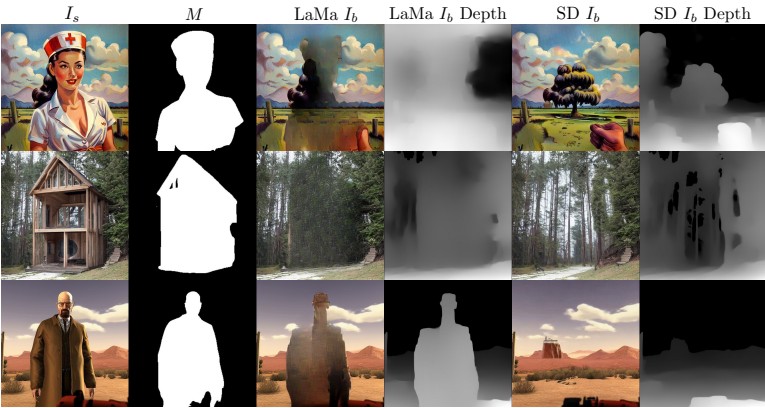

Figure 9: Comparison of LaMa and SD for inpainting, and its corresponding depth maps. Although images inpainted by LaMa seem to have their salient objects removed, their corresponding depth maps contain artifacts of the salient object.

**Inpainting of** $I_s \otimes (1 - \tilde{M})$. To generate $I_b$ in $\{I_f, I_b, M\}$, we utilize a variant of SD specifically trained for inpainting, setting the target prompt as "empty scenery, highly detailed, no people". We also test LaMa (Suvorov et al., 2022), a widely adopted inpainting model, and gauge its suitability as our inpainting module, focusing on the quality of the depth map of $I_b$. In Figure 9, we observe that the $I_b$ generated from LaMa exhibits certain artifacts that may not align well with the requirements of our pipeline. A notable characteristic of LaMa is that the depth maps of $I_b$ often retain the shape of the salient object, which could impact the information relayed to the local fuser. On the other hand, the SD inpainting module proves adept for the generation of $I_b$. Focusing on $I_b$ of the first row of Figure 9, it can be seen that certain objects that weren't once present in $I_s$ has been generated. This attribute of SD's inpainting module deems attractive to leverage in depth disentanglement training: to distill information about the relative placement of salient objects, it is critical for our model to effectively *see* objects occluded by the salient object during training. For a visualization on the synthetic image triplets generation pipeline, see Figure 8.

### A.3 ADDITIONAL RESULTS

**More Quantitative Results.** We provide additional quantitative results on the Pick-a-Pic validation set in Table 3. Following the trend on the COCO-Stuff validation set, our finetuned model excels over other models in all metrics with the exception of the FID and IS values from our depth-only experiment. Interestingly, when comparing results from the COCO-Stuff validation shown in Table 1, we observe that the performance rankings of each model remain largely consistent. However, the specific values for FID and IS metrics deteriorate significantly, while the CLIP Scores see notable improvement. One potential reason for this trend relates to the underlying nature of the pre-trained models, such as Inception-V3 (Szegedy et al., 2016), used in these metrics. While both sets of images being compared in this experiment are synthetic, these models are trained on real-world images, inherently capturing real-world image features and patterns. The synthetic nature of the Pick-a-Pic images might diverge considerably from these real-world expectations, thereby influenc-

| Model/Condition | Depth | | | Semantics | | | Depth+Semantics | | |
|---|---|---|---|---|---|---|---|---|---|
| | FID (↓) | IS (↑) | CLIP Score (↑) | FID (↓) | IS (↑) | CLIP Score (↑) | FID (↓) | IS(↑) | CLIP Score (↑) |
| GLIGEN | 22.540 | 12.733 | 28.227 | - | - | - | - | - | - |
| ControlNet | **21.183** | **13.685** | 28.112 | - | - | - | - | - | - |
| Uni-ControlNet | 24.561 | 13.260 | 28.053 | 28.964 | 12.809 | 25.245 | 22.808 | 12.006 | 26.722 |
| T2I-Adapter | 26.262 | 13.309 | 28.017 | 47.996 | 11.408 | 25.033 | 34.698 | 10.745 | 26.583 |
| CnC | 28.192 | 11.460 | 27.347 | 36.272 | 10.353 | 24.301 | 25.524 | 11.131 | 27.109 |
| CnC Finetuned | 32.155 | 11.512 | **28.274** | **26.042** | **12.838** | **27.681** | **22.484** | **12.602** | **28.094** |

Table 3: Evaluation metrics on the Pick-a-Pic val-set. We omit the results of semantics and depth+semantics on GLIGEN and ControlNet due to the models not supporting these conditions. Best results are in **bold**.

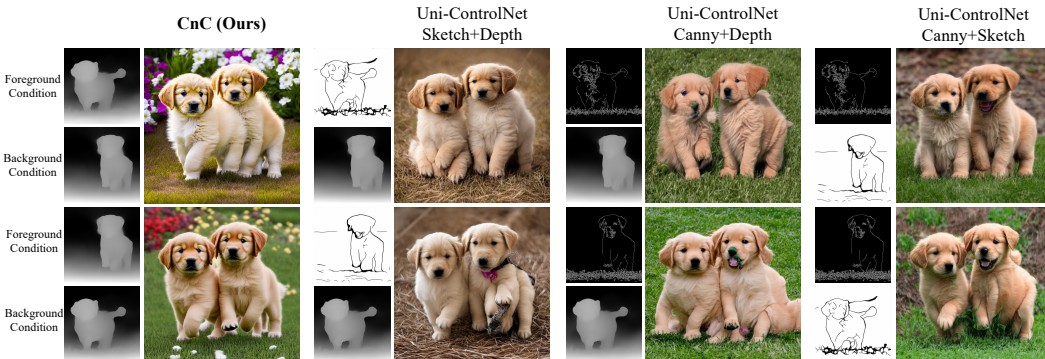

Figure 10: Comparing the ordering ability of localized conditions against Uni-ControlNet. Uni-ControlNet reports that the combinations listed are the most effective, against other combinations of 7 local conditions.

ing the FID scores. Moreover, even if both datasets under comparison are synthetic, the variance and distribution of features in the synthetic Pick-a-Pic dataset could be distinct enough from typical real-world datasets to lead to the observed differences in FID and IS scores. This highlights the nuances associated with evaluating models on synthetic versus real datasets and emphasizes the need for careful consideration when drawing conclusions from such evaluations.

**Ordering of localized objects.** In Figure 10, we compare our model's ability to place objects in front of another through the use of local conditions against Uni-ControlNet. Uni-ControlNet is able to take in 7 local conditions, and report that the local conditions pair listed in Figure 10 are the most robust in handling conflicting conditions, *i.e.* two overlapping objects. Although some samples do show the foreground local condition being placed in front of its counterpart, Uni-ControlNet often fails in conveying a sense of depth in its samples, resulting in the two objects to be generated on the same z-axis. On the other hand, even if the two depth maps conditioned have relatively same depth values, our model is able to consistently occlude overlapping parts of the background depth map.

**Additional details on Reconstruction.** In Figure 11, we provide additional samples of the depth maps extracted from the reconstruction experiment detailed in Section 4.2. Although depth maps produced by MiDaS does not reflect the true metric depth of an object, comparing the depth maps of the ground truth images and the reconstructed images shed light onto how well models reflects the images and depth maps exposed to them during training. While the reconstructed depth maps of baseline models hold the overall shape of an object, it can be seen that our model succeeds in capturing the relative depth of regions relative to the salient object. Additionally, we elucidate our choice in selecting MAE as a metric for gauging the quality of reconstruction. Although depth maps produced by MiDaS do not predict metric depth as mentioned above, our model and baseline models were trained to generate images based on image-depth map pairs. Comparing depth maps

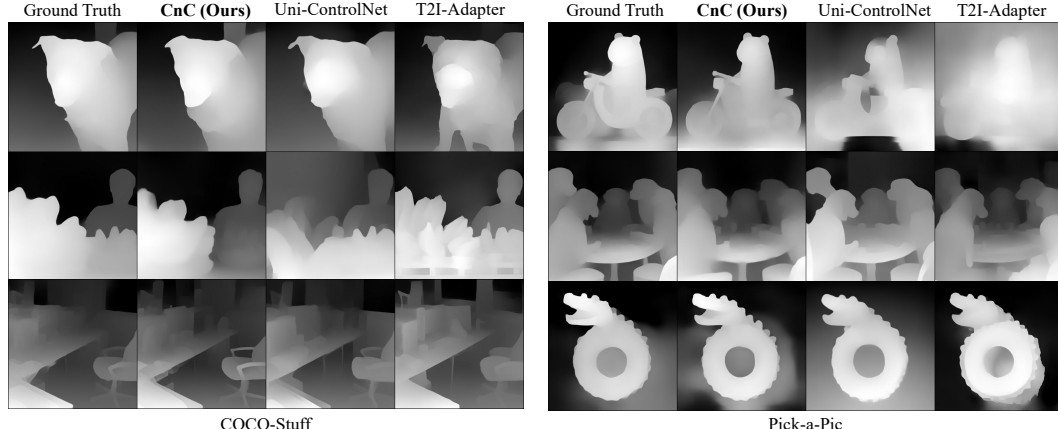

Figure 11: Qualitative comparison of depth maps extracted from reconstructed images.

| Model/Condition | COCO-Stuff | | | Pick-a-Pic | | |
|---|---|---|---|---|---|---|
| | FID (↓) | IS (↑) | CLIPScore (↑) | FID (↓) | IS (↑) | CLIPScore (↑) |
| Uni-ControlNet (Canny Edge) | **17.119** | **30.440** | 25.726 | 21.955 | **12.469** | 28.517 |
| T2I-Adapter (Canny Edge) | 20.051 | 28.449 | 25.850 | 30.547 | 12.230 | 28.412 |
| **CnC (Canny Edge, Ours)** | 17.745 | 29.809 | **26.283** | **20.501** | 12.215 | **28.786** |

Table 4: Canny edge evaluation metrics on the COCO-Stuff and Pick-a-Pic val-set. Best results are in **bold**.

can be thought of how well a model "predicts" depth maps given ground truth depth maps, which in turn gauges how well a model has learned the relationship between images and depth maps. In Figure 12, we visualize the relationship between how similar the ground truth depth maps are to the reconstructed depth maps in terms of MAE. Each set was randomly chosen from the top 50 pairs with the lowest/highest MAE values. It can be seen that the pairs with the lowest depth map MAE scores directly result in the quality of reconstruction, with the reconstructed images faithfully portraying the relative depth present in the ground truth images. On the other hand, the pairs with the highest MAE scores result in sub-par reconstructed images. Taking the second row of Figure 12(b) as an example, it can be seen that the reconstructed image fails in capturing the relative depth of the tree and person present in the ground truth image.

## A.4 ABLATION STUDY

**Ablation on different spatial conditions.** The ability of our model to effectively order objects into spatial regions stems from our depth disentanglement training, where the spatial information of, and what lies behind the salient object is distilled into their respective streams of the local fuser. To this end, it can be seen that our model can be trained on different types of local conditions, given that the condition holds spatial information of an image. We explore the capabilities of our local fuser, and show the effects of training on canny edges compared to depth maps in Figure 13. Canny edges hold properties and biases different to that of depth maps, in the way that canny edges values are binary, and trade off the ability to represent depth with more fine grained details. Because of these properties, it can be seen that while DDT does learn the relative placement of objects even with canny edges, using canny edges has its own pros and cons. Figure 13 (a) and (c) report cases where using depth maps are preferred, while (b) and (d) report the opposite. We find that canny edges often fail in generating a sense of depth, as seen in case (c) where the apples look relatively flat. However, this property can be preferred when leveraging base images that are flat to begin with. Figure 13(b) and (d) demonstrates such cases, where depth maps of flat base images (such as posters and vector graphics) fail to capture spatial information, resulting in sub-par images. We find that DDT is able to effectively leverage the inductive biases of a given representation, whether it be canny edges or depth maps, and special cases might call for variants of our model trained on

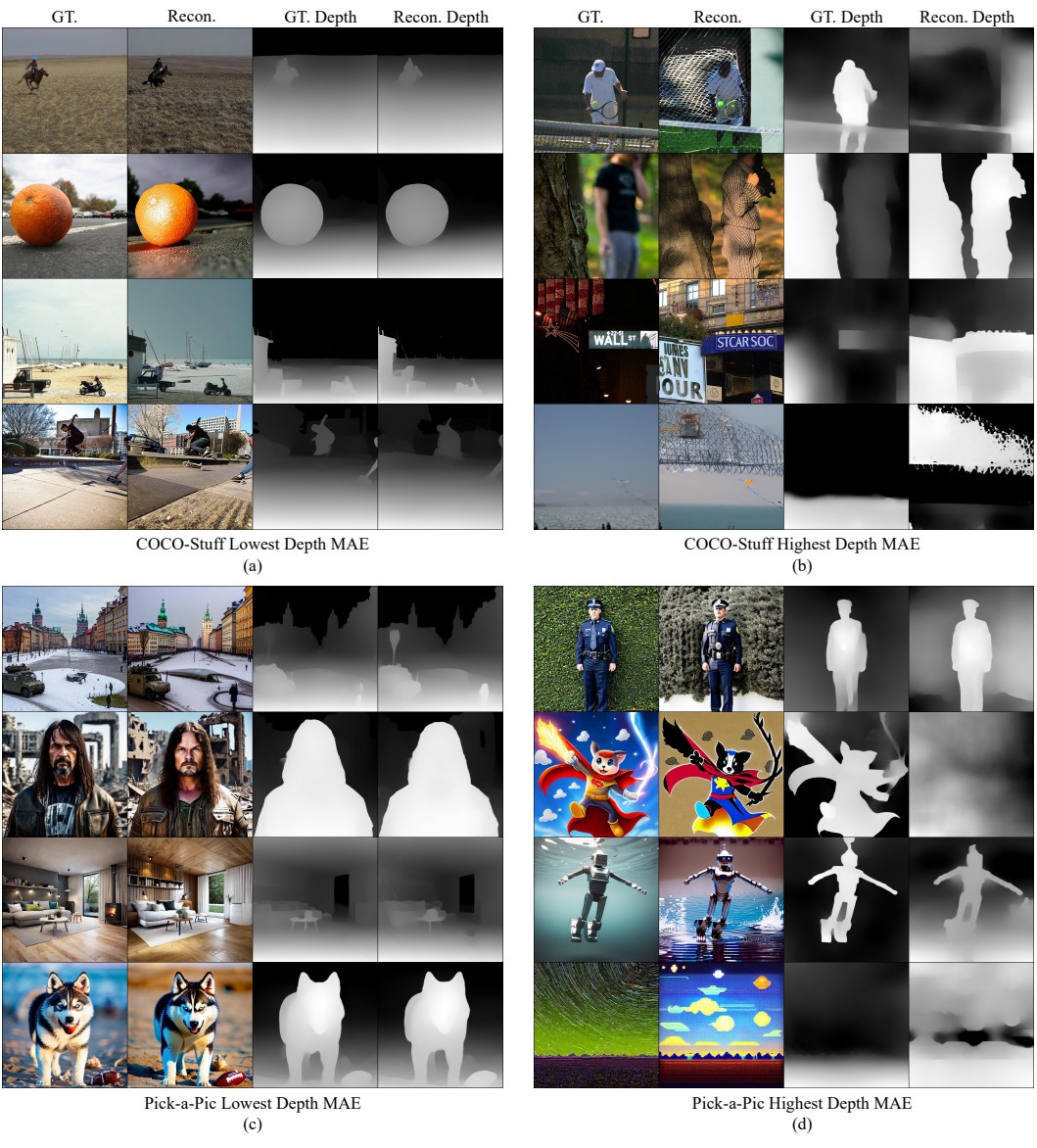

Figure 12: Qualitative comparison of depth map pairs with the lowest/highest MAE values from each dataset. Each set was randomly chosen from the top 50 pairs of the lowest/highest MAE values from COCO-Stuff and Pick-a-Pic, respectively.

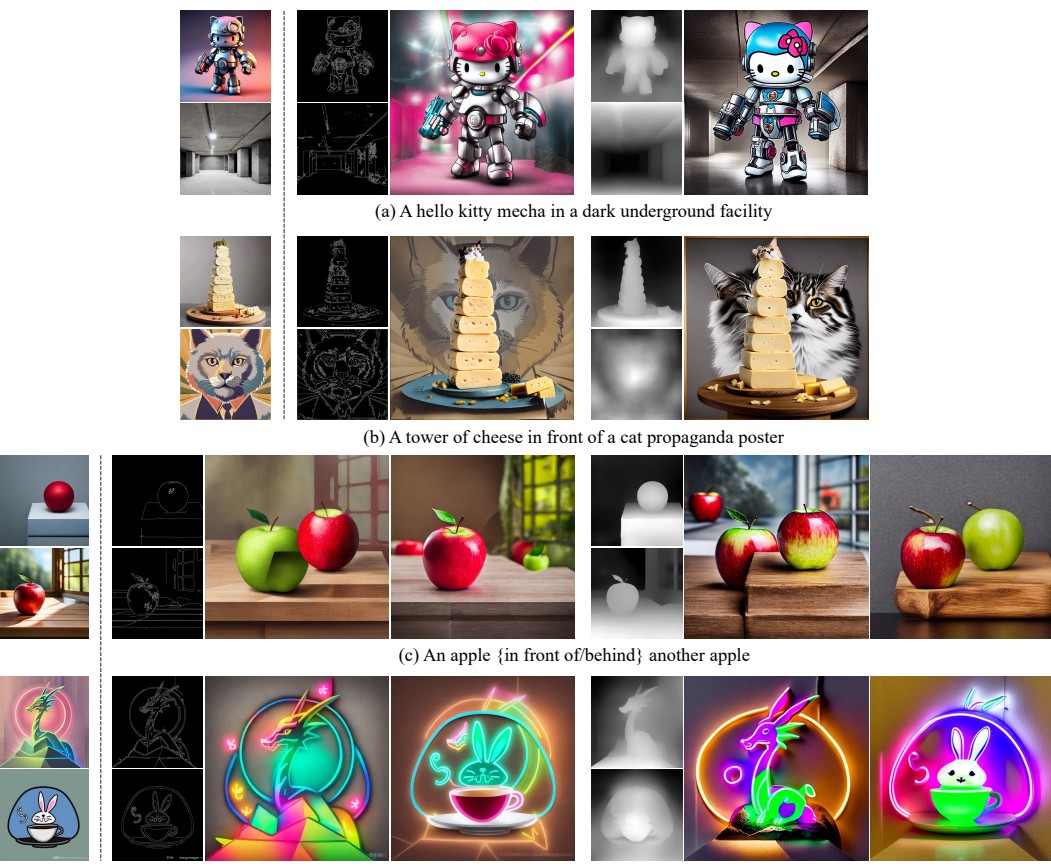

(a) A hello kitty mecha in a dark underground facility

(b) A tower of cheese in front of a cat propaganda poster

(c) An apple {in front of/behind} another apple

(d) A neon sign of a dragon {in front of/behind} a bunny in a cup

Figure 13: Ablation study on canny edge as a representation for our Local Fuser. (a) and (c) report cases where depth maps yield better images, while (b) and (d) report cases where canny edges might be preferred.

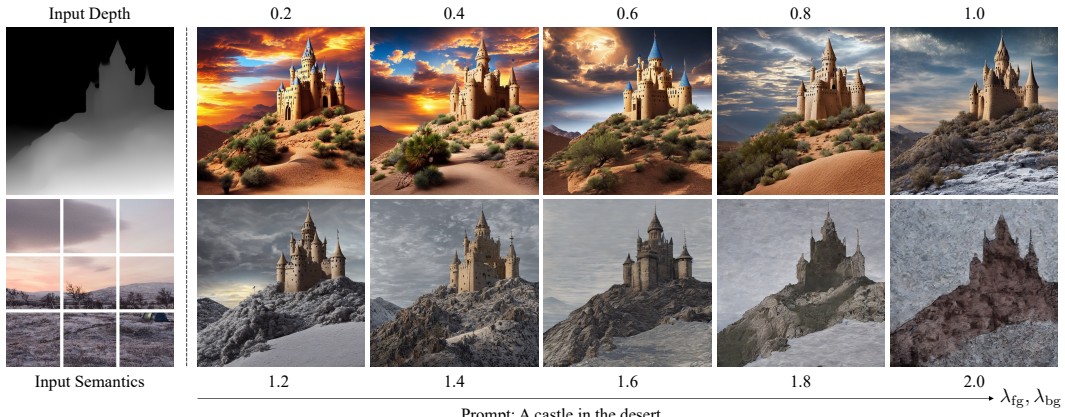

Figure 14: Conflicting text and semantics. We condition the same depth map and exemplar image to each foreground/background stream, and increase $\lambda_{fg}, \lambda_{bg}$ with the same increments.

different representations. We additionally report the quantitative results of our local fuser trained on canny edges, compared to other baseline models on the COCO-Stuff and Pick-a-Pic validation sets in Table 4. We follow the same experimental setup of our main quantitative experiments detailed in Section 4.2. We find the variant of our model trained on canny edges are comparable to other baseline models, resulting in the best CLIPScore for COCO-Stuff, and the best Inception Score and CLIPScore for Pick-a-Pic.

**Conflicting Text & Semantics.** We explore the use of conflicting text prompts and semantics, and the effect of hyperparameters $\lambda_{fg}$ and $\lambda_{bg}$ used to determine the influence of each global semantic. In Figure 14, same depth maps and exemplar images are conditioned to each foreground/background stream to generate each sample. We fix the text prompt to "A castle in the desert", which serves as a semantic contradiction to the exemplar image of an empty icy tundra. The effects of steadily increasing the hyperparameter values can be seen clearly, where the value of 1 strikes an adequate balance between the effect of the text prompt and the global semantic. Although the effect of the text prompt is nullified when the hyperparameters are set to larger values, it can be seen that the shape of the original object stays intact due to the combination of the depth map and soft guidance.

**Additional results on conflicting semantics.** Figure 15 shows the full effect of the hyperparameters $\lambda_{fg}$ and $\lambda_{bg}$ when conditioning two contradicting global semantics. The exemplar image for the foreground global semantic is an image of flowing lava, and the background global semantic is an image of a snowy field. By fixing the prompt with "A volcano" and increasing each hyperparameter $\lambda$, the effect of each semantic is steadily amplified. Empirically, we observe that setting both $\lambda_{fg}$ and $\lambda_{bg}$ to 1 yields results that are visually harmonious, which aligns with our training configuration where each hyperparameter is set to 1. We also provide a metric to gauge our model's ability to localize global semantics. Given a foreground and background exemplar image $I_{fg.sem}$, $I_{bg.sem}$, the corresponding generated image $I_{gen}$, and the mask utilized for soft guidance $M$, we leverage the CLIP image encoder to measure how well each semantic is applied to $I_{gen}$. The mask is utilized to create two images, $I_{gen} \otimes M$ and $I_{gen} \otimes (1 - M)$, and fed into the CLIP image encoder to create two image embeddings. These embeddings are then compared with the embeddings of the original exemplar images $I_{fg.sem}$ and $I_{bg.sem}$ via cosine similarity, and the average cosine similarity can be utilized as a metric of how well each semantic is applied into each region of $I_{gen}$. A high average cosine similarity implies that each semantic has been faithfully applied to each region via soft guidance. This characteristic can be seen in Figure 16, where two contradicting semantics of a hot air balloon and a coral reef, are localized. The average cosine similarity steadily increases as $\lambda_{bg}$ increases, which reflects how much the effect of the coral reef increases as can be seen in the generated images. The average cosine similarity drops sharply at $\lambda_{bg} = 2.0$, due to the model over saturating the effect of the coral reef, resulting in a low fidelity sample.

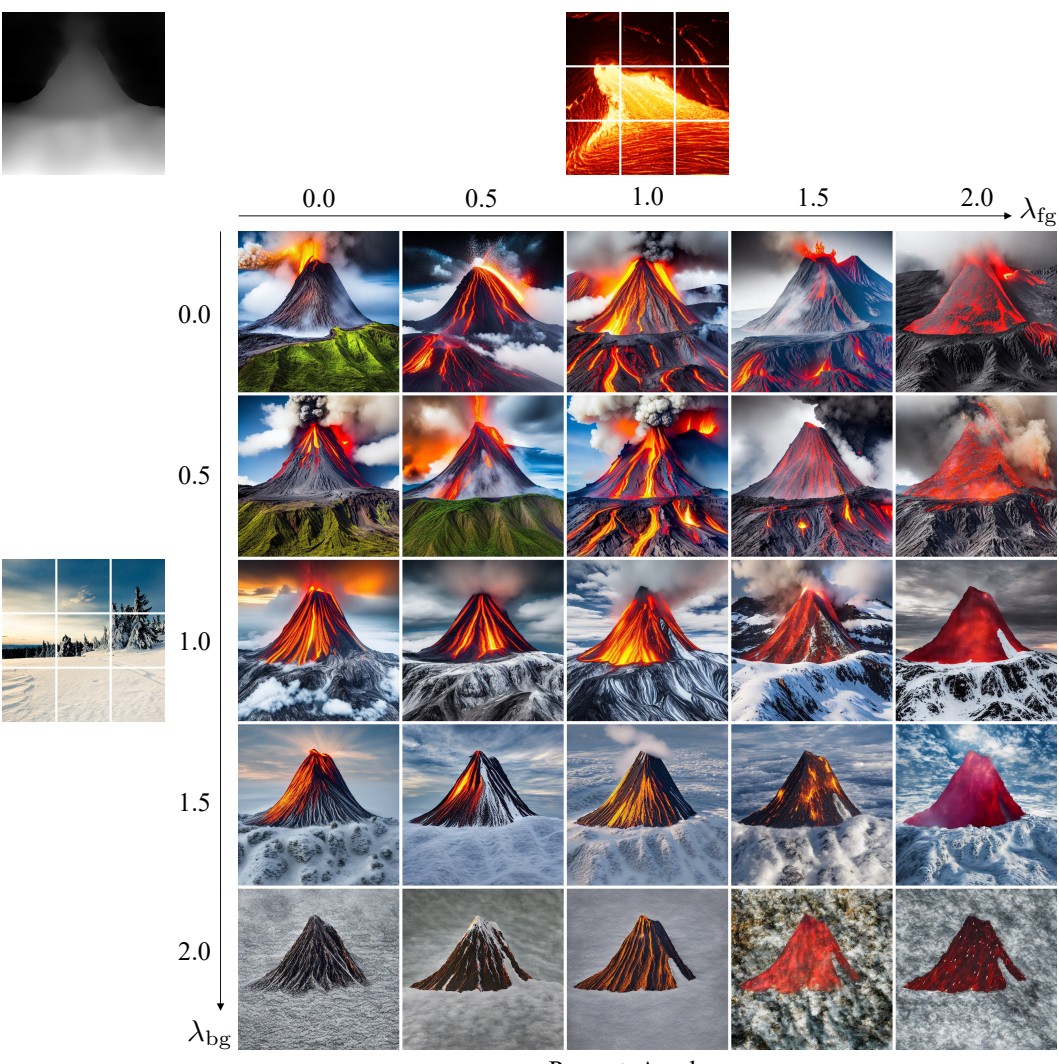

Prompt: A volcano

Figure 15: Conflicting semantics. The foreground global semantic image is an image of flowing lava, and the background global semantic image is an image of a snowy field. We fix the text prompt to "A volcano" and demonstrate the effects of hyperparameters $\lambda_{\text{fg}}$ and $\lambda_{\text{bg}}$.

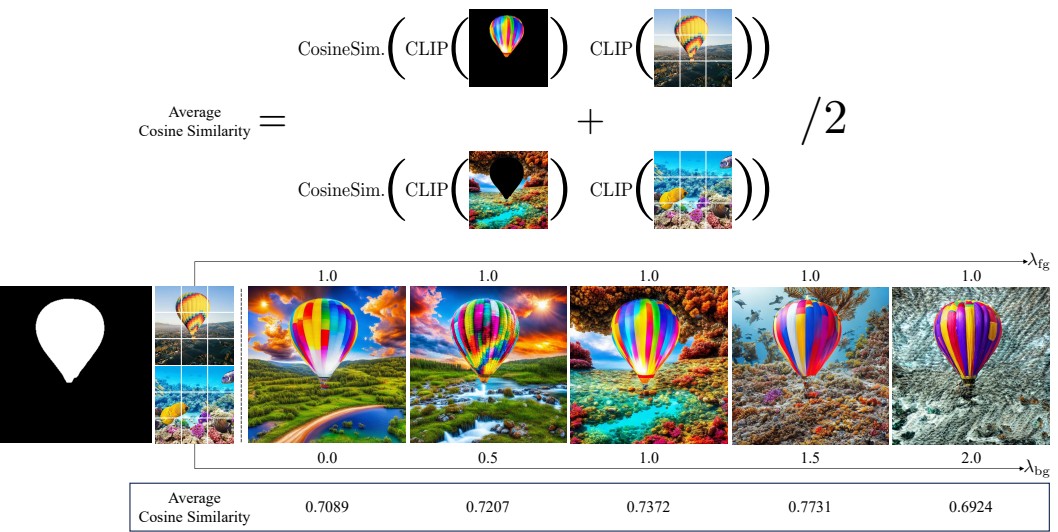

Figure 16: Additional results on the effect of soft guidance with conflicting semantics, and its respective average cosine similarity scores. We condition the same depth image for each stream in the local fuser, and generate samples without conditioning a prompt.

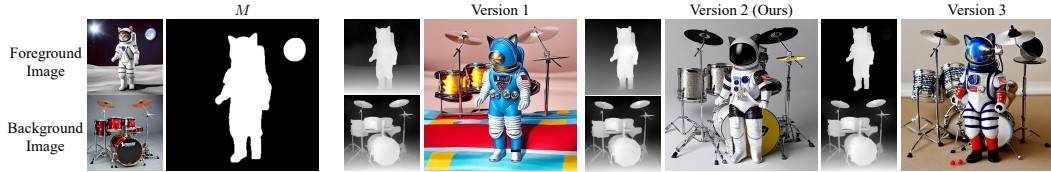

Figure 17: Depth map Ablations. We find that our model generalizes well to different versions of depth maps. Version 1 refers to depth maps extracted from $I_s$. Version 2 refers to depth maps extracted from $I_f$. Version 3 refers to depth maps extracted from $M \otimes \mathrm{depthmap}(I_f)$.

**Depth map Ablations.** The local fuser incorporates depth maps extracted from the synthetic image triplets $\{I_f, I_b, M\}$ during DDT. However, we find that our model is also able to generate samples conditioned on different *versions* of depth maps, in which we demonstrate in Figure 17. Version 1 refers to conditioning the foreground stream of the local fuser on the depth map extracted from $I_s$, instead of $I_f$. Version 2 refers to conditioning via how the local fuser was originally trained with DDT, or conditioning the depth map of $I_f$. Version 3 refers to conditioning the local fuser on $M \otimes \mathrm{depthmap}(I_f)$, or a masked version of the depth map of $I_f$. Pedagogically, the difference between version 2 and 3 can be thought of as whether masking with $M$ is done before or after the extraction of the depth map of $I_f$. Although our model is trained on depth maps of version 2, the sample generated by version 1 shows that our model has learned the relative positioning of objects through its interaction with the background depth map. Because the foreground depth map of version 1 retains additional depth cues of the ground, the generated sample retains this information and places parts of the drum set over the ground. Interestingly, we also find that version 3 also generates high fidelity samples, even though this form of depth maps are never found in the wild. This also can be attributed to the relative depth estimation capabilities of our model paired with the background depth map.

