# OpenReview forum: "Compose and Conquer: Diffusion-Based 3D Depth Aware Composable Image Synthesis"
_ICLR.cc/2024/Conference — ICLR 2024 poster_

### Official Review · Reviewer_FnjA · 2023-10-29

**Soundness:** 3 good
**Presentation:** 3 good
**Contribution:** 3 good
**Rating:** 6
**Confidence:** 3

**Summary:**

This article proposes two methods to develop a diffusion model that merges the capability to manipulate the three-dimensional positioning of objects with the application of disentangled global stylistic semantics from exemplar images onto the objects.

Specifically, the paper introduces depth disentanglement training, which makes the model realize the  3D relative positioning of multiple objects by disentangling the salient object depth and the background object depth for the fusion of the condition during training.  In the meantime, this work presents a technique called soft guidance, which imposes the mask information into cross-attention mechanism to facilitate apply global semantics onto targeted regions without specific local localization cues.

**Strengths:**

1. The paper is written in a clear and coherent style, presenting ideas in a manner that is easily comprehensible. Additionally, most figures in the paper effectively visualize and reinforce the concepts discussed.

2. As stated in the paper, this work is the first to leverage the disentaglement of images to salient object depth and impainted (unseen) background depth map for training the relative depth aware diffusion model.  The soft guidance technique is also novel for applying the global semantics to specific localizations.

3. The adequate experiment supports the effectiveness of the model. Both the qualitative and quantitative comparisons demonstrate that the model can control the relative placement of the objects and the effectively prevent the concept bleeding.

**Weaknesses:**

1. Correct me if I have misunderstood. I'm confused about the details of the soft guidance. I understand that the work wants to leverage the mask map to selectively impose the foreground embedding and background embedding for the cross-attention mechanism. However, I'm uncertain as to whether this process should take place during the computation of similarity or subsequent to it. I would appreciate it if the author could elucidate the specific dimensions and computational details related to both the cross-attention mechanism and the soft guidance technique, to enhance reader comprehension. Please refer to the questions for additional context on my confusion.

2. Apart from the standard metrics used for evaluating generative models, I wonder if there exist specific metrics that can accurately assess the model's capability to control the three-dimensional placement of objects and localize global semantics, as these are the primary objectives of this study. While the Mean Absolute Error (MAE) between the ground truth depth map and the depth maps derived from the generated images may offer some insight into the model’s proficiency in 3D object placement, I am curious about how we might effectively gauge its ability to localize global semantics. Could there be other metrics or methods of evaluation that address this second capability?

**Questions:**

1. In my understanding, the size of S is $i\times j$, where i is the number of queries, and j is the number of keys; the size of $W_K \dot y_{full}$ is $j \times C$ and the size of $W_Q \dot z_t$ is $ i \times C$. However, I am puzzled as to why $j$ needs to be greater than $2N$.

2. Additionally, I am uncertain about whether the mask should be applied along the dimension of $C$. It perplexes me that the mask is utilized on the calculated similarity rather than during the actual computation of similarity.

---

> ### Author Response · Authors · 2023-11-18
>
> We thank you for acknowledging the contributions of our paper, mainly:
> - Our proposed DDT is the first method to leverage image triplets to distill the relative depth positions of objects in a diffusion model.
> - Our proposed soft guidance is a novel method to localize global semantics.
> - Our experiments demonstrate the model's capabilities of relative placement of objects and its effectiveness in preventing concept bleeding.
>
> > [W1, Q1, Q2] Correct me if I have misunderstood. I'm confused about the details of the soft guidance. I understand that the work wants to leverage the mask map to selectively impose the foreground embedding and background embedding for the cross-attention mechanism. However, I'm uncertain as to whether this process should take place during the computation of similarity or subsequent to it. I would appreciate it if the author could elucidate the specific dimensions and computational details related to both the cross-attention mechanism and the soft guidance technique, to enhance reader comprehension. Please refer to the questions for additional context on my confusion.
> > 1. In my understanding, the size of $S$ is $i\times j$, where $i$ is the number of queries, and $j$ is the number of keys; the size of $W_K \cdot y_{full}$ is $j \times C$ and the size of $W_Q \cdot z_t$ is $i \times C$. However, I am puzzled as to why $j$ needs to be greater than $2N$.
> > 2. Additionally, I am uncertain about whether the mask should be applied along the dimension of $C$. It perplexes me that the mask is utilized on the calculated similarity rather than during the actual computation of similarity.
>
> [W1] Firstly, we would like to address your questions on soft guidance and why the mask is utilized on the calculated similarity rather than during the actual computation of similarity. Assuming “during the computation of similarity” means **before** $S$ is rescaled with softmax and multiplied with $V$ (eg. $S'=(QK^T/\sqrt{d})\otimes M'$), and “calculated similarity” means **after** S is rescaled with softmax and multiplied with $V$ (eg. $S'=\operatorname{softmax}(QK^T/\sqrt{d})\otimes M'$), we would like to clarify that soft guidance is **indeed applied during the computation of similarity, or before the softmax operation.** It seems that this confusion stems from us omitting the softmax operation in the paper, where we state that “…the attention operation $S' \cdot V$ is completed.”, and whereas we intended $S$ to be understood as the raw similarity scores, readers would assume that $S$ means the scaled similarity score (or attention score, depending on sources). We thank you for your detailed insight on this matter. We’ve updated the equations to be clear that soft guidance is indeed applied during the similarity computation, and the reshaped, flattened, repeated boolean mask $\varphi(M) \in \{0,1\}^{i \times N}$ to $\varphi(M) \in \mathbb{B}^{i \times N}$ to avoid confusion.
>
> [W1] Secondly, we would like to illuminate the specific dimensions used in the soft guidance technique. We bring up an exact example of how soft guidance is applied in the first cross-attention layer latents pass through, when generating an image of size $512 \times 512$.
> In our implementation, we use 8 attention heads, and set $N=4$, or the number of tokens $\mathbf{y}\_\text{fg}$, $\mathbf{y}\_\text{bg}$ are projected to. Since we use the CLIP text encoder, the given text prompt is encoded into $77$ tokens with a dimension of $768$. By $\mathbf{y}\_\text{full} = \operatorname{concat}(\mathbf{y}\_\text{text}, \lambda\_\text{fg}\mathbf{y}\_\text{fg},\lambda\_\text{bg}\mathbf{y}\_\text{bg})$, $\mathbf{y}\_\text{full}$ becomes a tensor of shape `[1, 85, 768]`. The latent $\mathbf{z}\_t$ is a tensor of shape `[1, 4096, 320]` because it has been processed by the subsequent self attention layer. $\mathbf{z}\_t$ and $\mathbf{y}\_\text{full}$ are then projected into $Q$, $K$, $V$, with respective dimensions of `Q=[8, 4096, 40]` , `K=[8, 85, 40]`, and `V=[8, 85, 40]`. The raw similarity scores are calculated by $S=QK^T/\sqrt{d}$, resulting in `S=[8, 4096, 85]`. We then reshape, flatten, repeat the mask $M$ used for soft guidance (with an initial dimension of `[1, 512, 512]`, to a shape of `fg_mask=[1, 4096, 4]`, which serves as the mask for the foreground token part of $\mathbf{y}\_\text{full}$. We then take the inverse of the mask, `bg_mask=[1, 4096, 4]`, as the mask for the background token part of $\mathbf{y}\_\text{full}$. We then create a base mask of only ones `base_mask=[1, 4096, 77]`, and concat the three masks to create $M'$ of shape `[1, 4096, 85]`. Then the operation of $S\otimes M'$ is conducted across all attention heads (eg. masking), to get $S'$. Then, the rest of the attention operation, $\operatorname{softmax}(S')\cdot V$ is performed.

---

> > ### Author Response · Authors · 2023-11-18
> >
> > [Q1, Q2] In your first question mentioning the dimensions of $Q$ with a size of $i \times C$, and $K$ with a size of $j \times C$, $C$ would equal to $40$, and it can be seen that the mask isn’t applied across $C$. Regarding to your question about why $j$ has to be greater than $2N$, this can be thought of as a notational choice since by concatenating $\mathbf{y}\_\text{text}, \mathbf{y}\_\text{fg},\mathbf{y}\_\text{bg}$, we expand the key space $j$, and the notion that $j$ has to be greater than $2N$ is used to define the all ones matrix $J \in \mathbf{1}^{i \times j-2N}$, which is the `base_mask=[1, 4096, 77]` mentioned above. We hope that this clarifies soft guidance, and we thank you for your detailed questions on its specifications. We appreciate your attention to this detail, and we’re glad to answer any additional questions.
> >
> >
> > > [W2] Apart from the standard metrics used for evaluating generative models, I wonder if there exist specific metrics that can accurately assess the model's capability to control the three-dimensional placement of objects and localize global semantics, as these are the primary objectives of this study. While the Mean Absolute Error (MAE) between the ground truth depth map and the depth maps derived from the generated images may offer some insight into the model’s proficiency in 3D object placement, I am curious about how we might effectively gauge its ability to localize global semantics. Could there be other metrics or methods of evaluation that address this second capability?
> >
> > [W2] We have added an ablation study on a metric that gauges our model's ability to localize global semantics in our revised paper, section [A.4] of our appendix. To provide a brief summary: given two exemplar images, a mask used during soft guidance, and the resulting generated images, we take the cosine similarity between the CLIP image embeddings of the masked generated image and its corresponding semantic, and take an average value. We've included an additional experiment in section [A.4], Additional results on conflicting semantics, that demonstrates how the average cosine similarity directly correlates with the generated image, in terms of how well each semantic is localized to their corresponding regions. If the average cosine similarity is high, it implies that each semantic in each region is very similar to its exemplar image. This metric can be thought of as a variant of CLIPScore, which is the cosine similarity between the CLIP image embedding of an image and the CLIP text embedding of a prompt.

---

> > > ### Comment · Reviewer_FnjA · 2023-11-23
> > >
> > > I appreciate the author's detailed response, they addressed my concerns.

---

> > > > ### Author Response · Authors · 2023-11-23
> > > > **Thank you!**
> > > >
> > > > We appreciate your efforts as your comments have made our paper more comprehensive. We will refine our paper to include the experiments for the final version. Thank you.

---

### Official Review · Reviewer_iDhm · 2023-10-30

**Soundness:** 3 good
**Presentation:** 3 good
**Contribution:** 3 good
**Rating:** 6
**Confidence:** 4

**Summary:**

This paper presents a method for controllable text-to-image generation. The method learns auxiliary modules on top of a pre-trained Stable diffusion model, and introduces a novel training scheme to facilitate compositional image synthesis given two depth images that represent the foreground and background. Further, foreground and background styles are controlled by separate images thanks to a localized cross-attention mechanism. The qualitative and quantitative experiments demonstrate that the proposed method outperforms several baselines in terms of image quality, image-text alignment and foreground-background disentanglement.

**Strengths:**

- The paper studies the composition of control signals in controllable text-to-image diffusion. Unlike previous approach which takes a single control image, the proposed method allows the conditioning on two depth images. This provides a means to separately control foreground and background image content. The method also enables localized control of image styles using exemplar images. To the best of my knowledge, compositional image generation remains a challenging problem, and this paper demonstrates one feasible solution to the problem by engineering pre-trained diffusion models.

- The paper presents a novel training scheme to instill depth awareness into the diffusion model. Training of the method relies on RGB images and their foreground / background depth maps which are not readily available. To this end, the paper introduces a simple strategy to create synthetic training data from single-view image datasets. This is key to the success of the proposed method and may be of interest to the image synthesis community in a broader context.

- The method allows localized control of image styles using exemplar images. The key idea is to limit the extent of cross-attention so that tokens representing an exemplar image only contribute to a local region in the image. Many works have used attention maps to localize objects or control their shapes and appearance. This paper for the first time uses attention maps to control (local) image styles.

- The experiments demonstrate superior qualitative and quantitative results. The proposed method outperforms several strong baselines in image quality and image-text alignment while supporting broader applications.

**Weaknesses:**

- Calling the model "depth-aware" is misleading. I would rather say it learns to compose two spatial layouts and generate a coherent image. Using the teaser figure as an example, the cat can appear either in front of or behind the cake given the same depth maps, and similarly, the castle can either occlude or be occluded by the mountain. In other words, the exact ordering of objects is not induced by the depth maps. This phenomenon is likely because the depth produced by MiDaS is scale and shift invariant (i.e., it is not metric depth, and background can appear closer than foreground).

- Since all that matters is generating a coherent image, I would imagine that other types of spatial conditions (e.g., segmentation masks for both foreground and background) can work equally well if used for training. I encourage the authors to test this hypothesis, and design additional ablation experiments to fully reveal the behavior of their model.

**Questions:**

- The illustration of soft guidance in Figure 2 is confusing. I personally prefer color coding of the attention maps to highlight the regions influenced by different tokens.

- Details about the reconstruction experiments (Figure 6) are lacking. It is unclear from the text what is the exact evaluation procedure. Also the MAE values reported in Table 2 is not meaningful, again because MiDaS does not predict metric depth. Please include qualitative comparison between the input and reconstructed depth maps.

---

> ### Author Response · Authors · 2023-11-18
>
> We thank you for acknowledging the main contributions of our work, stating that:
> - Our method poses as a feasible solution in the challenging problem of compositional image generation.
> - Our novel training scheme may be of interest to the community in a broader context.
> - Our experiments demonstrate superior qualitative and quantitative results.
>
> > [W1] Calling the model "depth-aware" is misleading. I would rather say it learns to compose two spatial layouts and generate a coherent image. Using the teaser figure as an example, the cat can appear either in front of or behind the cake given the same depth maps, and similarly, the castle can either occlude or be occluded by the mountain. In other words, the exact ordering of objects is not induced by the depth maps. This phenomenon is likely because the depth produced by MiDaS is scale and shift invariant (i.e., it is not metric depth, and background can appear closer than foreground).
>
> [W1] We agree on your insight stating that the exact ordering of objects are not induced by the depth maps. The ability to order objects into their successive foreground/background stems from the proposed depth disentanglement training (DDT), where information about the otherwise occluded background is retrieved via our image triplets and distilled into our local fuser. The depth maps produced by MiDaS are indeed scale/shift invariant, but they do hold attractive properties that lead us to choosing them as our local condition, instead of other representations (eg. segmentation maps). Because depth maps are single channel representation of an object's shape and holds the relative depth representation of an object in the scope of the image, they hold an inductive bias that we sought attractive to leverage. To summarize, the "depth-aware" property of our model doesn't stem from the depth maps themselves (although we do leverage its inductive biases), but its ability to process each depth map in their respective foreground/background stream in our local fuser and place them accordingly.
>
>
> > [W2] Since all that matters is generating a coherent image, I would imagine that other types of spatial conditions (e.g., segmentation masks for both foreground and background) can work equally well if used for training.
>
> [W2] We agree that different spatial conditions can work equally well during training. We have added an additional experiment that elucidates this case in our revised paper, section [A.4] Ablation Study, of the appendix. We explore the use of canny edges instead of depth maps, due to the fact that canny edges hold very different inductive biases to that of depth maps. Canny edges are inherently binary, and they hold much more fine grained detail about an object while trading off the ability to represent the relative depth of a pixel. The added figure 13 of page 18 showcases the results, where we find that our model generalizes well to canny edges. As mentioned above, this ability to generalize well stems from our depth disentanglement training, where information about the background is distilled to the model, even if the representation used doesn't hold inductive biases about the relative depth of an object. However, the inductive biases of each representation does come into play when generating images. In figure 13 (a) and (c), depth maps generate images that capture the sense of depth present in the conditioned images, while canny edges result in the generated images often looking flat. The opposite holds, for in figure 13 (b) and (d), it can be seen that canny edges generate better fidelity samples to that of depth maps when conditioned on flat, illustration based images. This leads to the conclusion that while DDT is successful in distilling information of what object to place in front of and cover, our model leverages the inductive biases of the representations used. If the representation inherently holds information about the relative depth of an image, DDT also distills this information into our model, resulting in the ability to compose images in a depth aware fashion.

---

> > ### Author Response · Authors · 2023-11-18
> >
> > > [Q1] The illustration of soft guidance in Figure 2 is confusing. I personally prefer color coding of the attention maps to highlight the regions influenced by different tokens.
> >
> > [Q1] We have updated Figure 2 in the revised paper, and we agree that color coding the influenced regions provide a much clearer description. Thank you for the recommendation!
> >
> >
> > > [Q2-1] Details about the reconstruction experiments (Figure 6) are lacking. It is unclear from the text what is the exact evaluation procedure. [Q2-2] Also the MAE values reported in Table 2 is not meaningful, again because MiDaS does not predict metric depth. [Q2-3] Please include qualitative comparison between the input and reconstructed depth maps.
> >
> > [Q2-1] The evaluation procedure of reconstruction experiments are as follows: given a ground truth image from the validation set, image triplets are constructed. Subsequently, background/foreground depth maps and CLIP image embeddings are extracted from the triplets, and are fed into our model as done during training. Since other baseline models do not accept either two depth maps or CLIP image embeddings, the depth maps and CLIP image embeddings from the ground truth images are extracted, and fed into said models. We thank you for pointing out the ambiguity, and we've added additional details to the revised paper, in section [4.2].
> >
> > [Q2-2] To address the choice of including MAE of depth maps for our reconstruction experiments: the depth maps produced by MiDaS indeed doesn't predict metric depth, but we've included the MAE values as a metric of how our model "predicts" depth maps (extracted from the reconstructed images) given the ground truth depth maps, since our model and other baseline models were trained on said MiDaS outputs. We've also included an experiment that further elucidates the choice of including MAE in our revised paper, in section [A.3] of our appendix. The experiment portrays pairs of ground truth/reconstructed images and their extracted depth maps, randomly chosen from the top 50 pairs of the lowest/highest MAE values. In the experiment, it can be seen that lower MAE values directly correlate to how close the ground truth and reconstructed images are, and higher MAE values result in unfaithful reconstruction, and an overall decrease in sample fidelity. Although we agree that MiDaS depth maps do not predict metric depth, we find it suitable in terms of a metric that gauges the quality of reconstruction.
> >
> > [Q2-3] We've included a qualitative comparison figure between the input and reconstructed depth maps in our revised paper, in section [A.3], additional details on Reconstruction of our appendix. The figure shows that while other models hold the overall shape of the objects, the depth maps of our model succeeds in capturing the relative depth of areas relative to the salient object. We thank you for your recommendation, for the added comparison sheds light on how the actual ground truth depth maps and reconstructed depth maps interact.

---

> > > ### Comment · Reviewer_iDhm · 2023-11-23
> > >
> > > Thanks for addressing my comments. The revised draft looks good to me, and I would like to keep my rating.

---

> > > > ### Author Response · Authors · 2023-11-23
> > > > **Thank you!**
> > > >
> > > > We appreciate your contributions which have enriched the quality of our paper. Thank you, and we'll make sure the final manuscript is well revised to hold the newfound experiments.

---

### Official Review · Reviewer_taqR · 2023-11-08

**Soundness:** 3 good
**Presentation:** 3 good
**Contribution:** 3 good
**Rating:** 8
**Confidence:** 4

**Summary:**

The authors propose the Compose and Conquer (CnC) network that achieves 3D object placement and successfully integrates global styles and local conditions. To this end, the authors first propose depth disentanglement training (DDT) which disentangles the foreground and background depth and processes them with independent layers before fusing them together. Moreover, the paper also involves a novel soft guidance block that efficiently combines global and local conditions. Thorough qualitative and quantitative evaluations demonstrate the design and the effectiveness of the proposed method.

**Strengths:**

The strengths of the proposed paper can be summarized as:
1. The paper is well-written and easy to follow
2. The proposed DDT and soft guidance modules are novel and effective, demonstrated by both qualitative and quantitative results
3. Evaluations are comprehensive and showcase better results than existing SOTA methods

**Weaknesses:**

The weaknesses of the proposed paper can be summarized as:
1. Type of conditions. (1) Is the model capable of applying different types of conditions? (2) Is the model capable of applying two different conditions simultaneously while recognizing the 3D relations? I am rather interested in these different situations especially considering that the authors only employ depth in the submitted paper. More examples or scenarios would be appreciated.
2. Image triplets. There is no visualization of the prepared image triplets for training. I am curious regarding the quality of foreground image, background image and foreground mask. It's also especially important to analyze the inpainted background image and how it would negatively affect the training and final outcomes.
3. Qualitative results. (1) What are the prompts for examples in Figure 3? (2) Through the visualization in Figure 3, 4 and 5, it's interesting to see that the final generated images do not fully reflect the foreground depth condition. Meanwhile, the background depth map is often ignored through the qualitative results.
4. No limitations and societal impacts are discussed in the submission.

**Questions:**

N/A

---

> ### Author Response · Authors · 2023-11-18
>
> We thank you for acknowledging the strengths of our paper in that:
> - The methods we propose, DDT and soft guidance are novel and effective in enhancing the capabilities of diffusion models.
> - The evaluations are comprehensive and quantitative results showcase better results than existing SOTA methods.
>
> > [W1-1] Type of conditions. Is the model capable of applying different types of conditions? [W1-2] Is the model capable of applying two different conditions simultaneously while recognizing the 3D relations?
>
> [W1-1] Yes, our model indeed is capable of applying different types of conditions. We have added a section in our revised paper, section [A.4] Ablation Study of the appendix. We explore the usage of canny edges as an alternative to depth maps, due to the fact that while canny edges are spatial representations of an image, it trades off the ability to represent depth with more fine grained details. We include qualitative samples comparing generated images from canny edges and depth maps, in Figure 13. It can be seen that our model generalizes well to canny edges, a trait that stems from our depth disentanglement training where information about the salient object, and what's behind said object is distilled into our local fuser. However, canny edges hold inherently different inductive biases to that of depth maps. This can be seen playing a role in the generated samples, where canny edges yield better results for "flat" images (such as  graphics), but often fail in capturing a sense of depth. The opposite holds for depth maps, so we find that special cases might call for variants of our model trained on different conditions. We have also included a quantitative evaluation in the same section, where we compare the canny edge variant of our model with other baseline models, and find the results to be comparable. [W1-2] Because our model generalizes well to different conditions, we find no reason for our model to fail in generalizing different conditions for the foreground and background. While due to limitations in our computational resources at the moment, we plan to explore this possibility, starting with canny edges for the foreground and depth maps for the background. This initial choice comes from our newly found insights while experimenting with the canny edge variant of our model, where we find that canny edges provide more control over the intricate details of an object. We hypothesize that leveraging depth maps as background representations with canny edges as foreground representations might give us the best of both worlds; fine grained control over the salient object while additionally providing a sense of depth.
>
>
> > [W2] Image triplets. There is no visualization of the prepared image triplets for training. I am curious regarding the quality of foreground image, background image and foreground mask. It's also especially important to analyze the inpainted background image and how it would negatively affect the training and final outcomes.
>
> [W2] We kindly refer the reviewer to section [A.2] of the appendix of our paper, where we detail the process and provide visualizations of preparing the image triplets. We agree that the quality of the synthetic image triplets does influence the quality of the outcomes, especially the background image. Figure 9 of section [A.2], Details on CnC, of our appendix elucidates this factor about the image triplets, where we initially try out [1]LaMa, a widely utilized inpainting model. We show that the depth maps of the background images extracted by LaMa holds artifacts that don't provide information of what is behind the salient object, whereas background images extracted by SD don't. For depth disentanglement training to be fully effective, the background image has to distill information that is initially occluded by the salient object. If the quality of the background image were to be on par with the samples extracted by LaMa as in figure 9 (i.e. doesn't provide any information about the initially occluded background), it would hold the same effect of training our model with the same depth maps, as shown in figure 3 of our paper.
>
> We also find the process of binary dilation to be crucial in inpainting with SD, because the masks $M$ extracted by the Salient Object Detection model we utilize are usually pixel-perfect. SD's inpainting module tends to leave certain edge artifacts when the masks just barely cover the targeted object, so we additionally use binary dilation to expand the mask. These extra steps while utilizing a much computationally demanding module (SD over LaMA) are taken to ensure the quality of our image triplets, for without it would negatively effect the quality of final outcomes, as you've pointed out.
>
> [1]Suvorov, Roman, et al. "Resolution-robust large mask inpainting with fourier convolutions." _Proceedings of the IEEE/CVF winter conference on applications of computer vision_. 2022.

---

> > ### Author Response · Authors · 2023-11-18
> >
> > > [W3] Qualitative results. [W3-1] What are the prompts for examples in Figure 3? [W3-2] Through the visualization in Figure 3, 4 and 5, it's interesting to see that the final generated images do not fully reflect the foreground depth condition. Meanwhile, the background depth map is often ignored through the qualitative results.
> >
> > [W3-1] The examples in figure 3 are generated from the validation set of Pick-a-Pic and COCO-Stuff, from which the prompts were utilized. The prompts are:
> > 3(a) - A beautiful dessert waiting to be shared by two people (COCO-Stuff)
> > 3(b) - diamond minecraft block facing frontwards on a green background (Pick-a-Pic)
> > 3(c) - cat knight, portrait, finely detailed armor, intricate design, silver, silk, cinematic lighting, 4k (Pick-a-Pic)
> > We have added the prompts to the revised paper, in figure 3. Thank you for highlighting this aspect.
> >
> > [W3-2] A Depth map can be considered to be reflected in a generated image if the local placements (x, y axis) are reflected. In our case, we also consider the z-axis, so a depth map can be considered to be reflected in a generated image if the foreground depth map effectively occludes the background depth map in regions the foreground depth map is present. Figure 3 and 5 demonstrates these capabilities, and in figure 4, we only visualize a single depth map since the baseline models we compare our samples to only accept a single depth map condition. As detailed in section [4.2] of our paper, we condition the same depth map for our foreground/background streams, hence the visualization of a single depth map in figure 4.
> >
> >
> > > [W4] No limitations and societal impacts are discussed in the submission.
> >
> > [W4] We kindly refer the reviewer to page 9, section [5] Conclusion of our paper, where we discuss the limitations of our work in that the current framework limits the number of available conditions to one for each foreground/background stream. We have also updated the revised paper to include additional limitations, where the spatial disentanglement is limited only to the foreground and background, and we leave  disentangling the middle ground for future work. We also like to refer the reviewer to page 10 of our paper, where we include an ethics statement, discussing the societal impacts of our work.

---

> ### Comment · Reviewer_taqR · 2023-11-22
>
> I appreciate the authors' detailed responses, which effectively addressed most of my preliminary concerns. Hence, I would like to raise my rating. The authors may want to carefully revise the manuscript and incorporate the materials in the rebuttal.

---

> > ### Author Response · Authors · 2023-11-22
> > **Thank you!**
> >
> > We thank you for your detailed feedback and efforts, and we're grateful for your help in making our paper more robust. We will make sure to revise the manuscript for the final version.

---

### Author Response · Authors · 2023-11-21

We thank all reviewers for their thoughtful insights and constructive feedback on our work. We have carefully revised each and every feedback, and have updated our paper in order to reflect the given suggestions, and improve the clarity and depth of our work. Newly added and updated sections are highlighted in blue. We briefly summarize the revisions made below:

**Results of training on different spatial conditions (`reviewer taqR, iDhm`).** We have added an additional section in `page 16, A.4 Ablation Study-Ablation on different spatial conditions` that showcases our model trained on canny edges, instead of depth maps. The experiment shows that our model indeed is able to process different conditions, an ability enforced by our proposed depth disentanglement training. We show that while canny edges and depth maps hold different inductive biases, our model is able to capture each inductive bias while staying faithful to each foreground/background stream.

**Details and additional figures on reconstruction (`reviewer iDhm`).** We have added an additional section and 2 figures in `page 15, A.3 Additional Results-Additional details on Reconstruction`. This additional section provides insights on the relationship between the ground truth depth maps and the reconstructed depth maps, and also elucidates our choice on selecting MAE as a metric for the structural similarity of reconstructed samples.

**Metric for gauging the ability to localize global semantics (`reviewer FnjA`).** In `page 19, A.4 Ablation Study-Additional results on conflicting semantics`, we have included another sample that demonstrates the effect of soft guidance, and a metric that evaluates how well a semantic has been localized.

**Update for Figure 2 (`reviewer iDhm`).** We've added color coding to `Figure 2` for soft guidance to improve figure comprehension.

**Update for Figure 3 (`reviewer taqR`).** We have added the prompts that were used to generate samples in `Figure 3`.

**Added details to the reconstruction experiment (`reviewer iDhm`).** In `page 8, 4.2 Evaluation-Reconstruction`, we've added additional details on the exact process utilized for our reconstruction experiment.

**Updates for Soft Guidance related equations (`reviewer FnjA`).** We have made changes to the equations in `page 5, 3.3 Global Fuser - Soft Guidance` and `page 13, A.2 Details on CnC - Details on the Global Fuser` to enhance reader comprehension.

**Update to Conclusions (`reviewer taqR`).** We have renamed `page 9, 5 Conclusion` to `Conclusion & Limitations`, and added limitations of our model that we plan to address in the future.

---

### Meta-Review · Area_Chair_GLGv · 2023-12-05

**Metareview:**

This paper proposes a method to generate 2D images conditioned on depth maps. Specifically, the method includes a local fuser, a global fuser and cloned encoder/decoder from a Stable Diffusion model. The main contributions include: a) a novel dataset synthesis method that produces foreground/background/mask triplet, b) a depth disentanglement training schema multi-resolution features and c) a soft guidance module to inject style from guidance images. Reviewers recognize the paper as novel and effective. The major concern is the generalization ability of the proposed method to other conditions, which the authors addressed by training the model on canny edge images during the rebuttal.

**Justification For Why Not Higher Score:**

The paper targets at an interesting task -- compositional image generation conditioned on depth images. It shows impressive and realistic results. However, this task has limited scope and impact on the vision research community, thus I do not recommend it for spotlight or oral representation.

**Justification For Why Not Lower Score:**

All reviewers agree this paper proposes an effective method for a novel task and vote for acceptance. The major concern about the method generalization ability has been addressed during rebuttal. Other questions on implementation details have also been sufficiently resolved. Thus I recommend for acceptance.

---

### Decision · Program_Chairs · 2024-01-16

Accept (poster)